# Enhanced net $CO_2$ exchange of a semideciduous forest in the southern Amazon due to diffuse radiation from biomass burning

Simone Rodrigues[1,*], Glauber Cirino[1,4,5,*], Demerval Moreira[2], Andreae Pozzer[3], Rafael Palácios[4,5], Sung-Ching Lee[6], Breno Imbiriba[4], José Nogueira[†], Maria Isabel Vitorino[1,4], and George Vourlitis[7,*]

[1]Programa de Pós-Graduação em Ciências Ambientais, Universidade Federal do Pará, Belém-PA, Brazil
[2]Faculdade de Ciências, Universidade Estadual Paulista, Bauru-SP, Brazil
[3]Atmospheric Chemistry Department, Max Planck Institute for Chemistry, Mainz, Germany
[4]Instituto de Geociências, Faculdade de Meteorologia, Universidade Federal do Pará, Belém-PA, Brazil
[5]Programa de Pós-Graduação em Gestão de Risco e Desastre na Amazônia, Universidade Federal do Pará, Belém-PA, Brazil
[6]Department Biogeochemical Integration, Max Planck Institute for Biogeochemistry, Jena, Germany
[7]Department of Biological Sciences, California State University, San Marcos, CA, USA
[*]These authors contributed equally to this work
[†]Deceased
**Correspondence:** G. Vourlitis (georgev@csusm.edu)

**Abstract.** Carbon cycling in the Amazon fundamentally depends on the functioning of ecosystems and atmospheric dynamics, which are highly intricate. Few studies have hitherto investigated or measured the radiative effects of aerosols on the Amazon and Cerrado. This study examines the effects of atmospheric aerosols on solar radiation and their effects on Net Ecosystem Exchange ($NEE$) in an area of semideciduous tropical forest in the North of Mato Grosso State. Our results show that for a relative irradiance ($f$) 1.10-0.67, a decrease in incident solar radiation is associated with a reduction in the NEE. However, an average increase of 35-70% in $NEE$ was observed when pollution levels Aerosol Optical Depth (AOD) were above $\approx 1.25$ and $f < 0.5$. The increase $NEE$ was attributed to the increase of up to 60% in the diffuse fraction of Photosynthetically Active Radiation. These results were mainly attributable to the biomass burning organic aerosols from fires. Important influences on Vapor Pressure Deficit ($VPD$) and air Temperature ($T_{air}$) and Canopy ($LC_T$), induced by the interaction between solar radiation and high aerosol load in the observation area, were also noticed. On average, a cooling of about 3-4 °C and a decrease up to 2-3 hPa was observed for $T_{air}$, $LC_T$, and $VPD$, respectively. Given the long-distance transport of aerosols emitted by burning biomass, significant changes in atmospheric optical properties and irradiance will impact the $CO_2$ flux of semideciduous forests distributed in the region.

## 1 Introduction

Carbon (C) is a key element in global biogeochemical cycles, and understanding the biosphere-atmosphere fluxes of mass and energy is essential to understanding current and future terrestrial C storage. The role of Amazonian Forest ecosystems has been widely debated (Booth et al., 2012; Huntingford et al., 2013; Brienen et al., 2015), especially for Amazonian tropical forests (Doughty et al., 2015; Gatti et al., 2014, 2021). Redistribution of biomes and plant species (Davison et al., 2021), loss of

biodiversity (Brando et al., 2014; Saatchi et al., 2021), increase in fires (Brando et al., 2019; Alencar et al., 2022; Sullivan et al., 2020), outbreaks of pests and diseases (Anderegg et al., 2020) are examples of impacts, aggravated not only by climatic factors but also by anthropogenic ones (Ometto et al., 2022). These impacts have been threatening the largest pantropical $CO_2$ sinks since 1990'. Reductions from 1.26 PgC yr$^{-1}$ to 0.29 PgC yr$^{-1}$ are expected between 1990-2030, possibly reaching zero in the Amazon (Hubau et al., 2020). The result of increasing atmospheric $CO_2$ levels provides important feedback on the future of greenhouse warming (Booth et al., 2012; Huntingford et al., 2013). In the Amazon biome, forest ecosystems play an important role in terrestrial C storage, and while these forests seem to have a uniform behavior, there are distinct climatic sub-regions that affect C storage (Brienen et al., 2015; Gatti et al., 2021). $CO_2$ absorption through photosynthesis increases the vegetation and soil C stocks, representing a C sink, while plants, animals, microbial respiration, decomposition of dead vegetal biomass, and wildfires release $CO_2$, representing a C source to the atmosphere (Artaxo et al., 2022; M. Venturini et al., 2023; Junior et al., 2020).

In general, the participation of forests in the global carbon cycle can only be adequately quantified by long-term studies monitoring C exchange at the plant-atmosphere interface. Forests are estimated to store 200-300 Pg C (Pan, 2011; Saatchi et al., 2011; Avitabile et al., 2016), about a third of what is contained in the atmosphere. This stock is very dynamic, and these trees process about 60% of global photosynthesis, sequestering about 72 Pg C from the atmospheric component through Gross Primary Production ($GPP$) every year (Beer et al., 2010) but releasing a similar amount back into the atmosphere from ecosystem (plant +animal+microbial) respiration (Nagy et al., 2018). With these large fluxes, a small proportionate change in $CO_2$ uptake or release can result in a large net C storage or sink. Carbon concentrations in the atmosphere have increased since the beginning of the industrial period and currently act with other C emission sources, such as the degradation of forests, mainly tropical ones. Recent reports (Gatti et al., 2021) show that some regions of the Amazon act as a source of $CO_2$ to the atmosphere as a result of logging, land use change, and fires that occur in the region. However, regional numeric modeling (Moreira et al., 2017) and in-situ studies indicate (Carswell et al., 2002; von Randow et al., 2004) that Amazonian forests can occasionally be net atmospheric $CO_2$ sinks; or approximately in equilibrium (Vourlitis et al., 2011). In general, the balance between rates of carbon emission or carbon fixation is delicate, so small external disturbances can change the dynamics of the forest and the state of the climate system.

Among the modulating agents of the $CO_2$ balance, solar radiation is a fundamental component for both photosynthesis and respiration. In Brazil, and especially in the Amazon region, the biomass burning emits large amounts of gases and aerosols into the atmosphere, which can strongly alter radiative fluxes, impacting $CO_2$ flux (Aragão et al., 2018; Malavelle et al., 2019; Morgan et al., 2019; de Magalhães et al., 2019). Atmospheric aerosols from biomass burning affect ecosystem light use efficiency ($LUE$) and productivity, influence the amount and nature of solar radiation received in the system, and affect other environmental conditions such as temperature and humidity (Kanniah et al., 2012; Mercado et al., 2009). Studies of the effects of aerosols on terrestrial C cycling processes have found positive, negative, and neutral effects, and most of the research in the Amazon have been conducted in the central (Cirino et al., 2014), eastern (Doughty et al., 2010; Oliveira et al., 2007), and southwestern (Yamasoe et al., 2006; Cirino et al., 2014) parts of the basin. However, little research has been done on the ecotones in the Amazon, e.g., in the Cerrado-Amazonian Forest transition, which lies within the arc of deforestation, and other

biomes such as Cerrado-Caatinga, Cerrado-Atlantic Forests, and Pantanal forests. Numeric simulations have also demonstrated the impact of aerosols on $GPP$ on the regional (Moreira et al., 2013; Rap, 2015; Bian et al., 2021) and global scales (Mercado et al., 2009; Rap et al., 2018), but physical representations of these impacts on transition ecosystems are still lacking.

The models, however, need improvements in parameterizing the radiative effects of aerosols and clouds on the $NEE$, e.g., a more realistic representation of the canopy structure and leaf physiological and morphological processes (Durand et al., 2021). Improvements in the aerosol optical model, its properties, secondary formation, lifetime, evolution, and absorption of aerosols are also critical (Drugé et al., 2022), especially those related to shape, size, and chemical composition. These improvements are fundamental for a more accurate and realistic spatial distribution of the atmospheric $CO_2$ absorption potential by Amazonian forests (Procopio et al., 2004; Moreira et al., 2017). In this sense, the potential for fire-induced atmospheric aerosols to impact to $CO_2$ absorption by tropical semideciduous (seasonal) forests in Mato Grosso (in the arc of deforestation) has not been evaluated either by direct observation or numerical modeling. It is known that these forests play a central role in preserving biodiversity (Fu et al., 2018), are located on the frontier of deforestation, and experience seasonal variations in $NEE$ (Vourlitis et al., 2011). These attributes make this region an excellent laboratory to assess the effects of atmospheric aerosols on forest $NEE$.

This research focuses on studying the action of biomass burning aerosols in an area of semideciduous forest located in the southern portion of the Amazon Basin, in the region the arc of deforestation of northern Mato Grosso, Brazil. To this end, we specifically seek to: (1) develop a clear-sky irradiance algorithm using a long observation period of Aerosol Optical Depth (AOD); (2) quantify the increase in the diffuse fraction of solar radiation due to the presence of aerosols from fires in the experimental study area; (3) quantify net and relative changes in $NEE$ from changes in direct and diffuse radiation; and to (4) evaluate the influence of fires on biophysical variables that influence forest photosynthetic rates, such as leaf canopy temperature ($LC_T$), air temperature ($T_{air}$), and the vapor pressure deficit ($VPD$). Aerosol data and micrometeorological measurements with carbon fluxes measured by the eddy covariance system were used from 2005 to 2008. All solar radiation measurements are evaluated in terms of aerosol depth (AOD), solar zenith angle ($SZA$), and relative irradiance ($f$). To our knowledge, this is the first study with this purpose.

## 2 Materials and Methods

### 2.1 Site descriptions

The study area was located in the south of the Amazon basin, 50 km northeast of Sinop, in the municipality of Cláudia (Lat 11° 24.75' S, Long. 55° 19.50' W), in the State of Mato Grosso (Fig. 1). This forest is located in the arc of deforestation, a region of continuous agricultural expansion (areas for soybean and cattle pasture, logging, and fires) (Barbosa et al., 2023; Nepstad et al., 2014; Balch et al., 2015; Alencar et al., 2022) (Figs. S1, S2, and S3), and is recognized as seasonal, dry, or semideciduous forest (Ackerly et al., 1989; Ratter et al., 1978). Supplementary Figs. S3 and S4 show the study area different aerosol loads during the dry and rainy seasons, respectively. These figures were obtained from the time series of the Terra and Aqua Satellites (AOD$_m$, Table1).

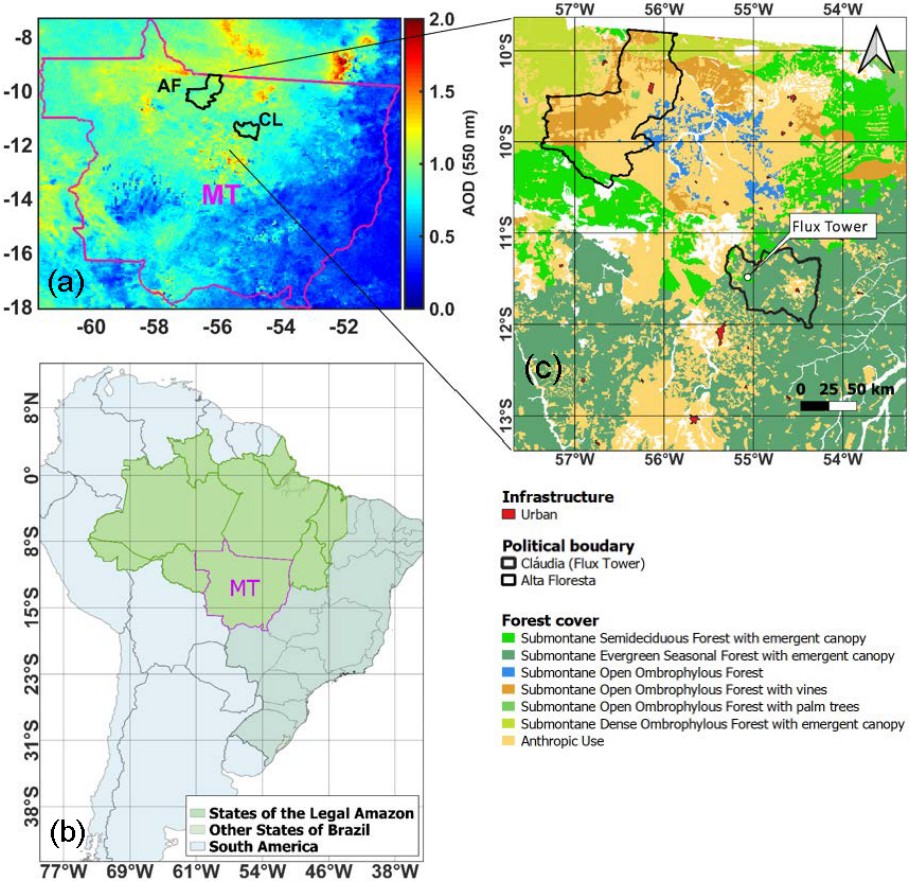

**Figure 1.** (a) shows the average regional distribution of AOD at 550 nm extracted from Terra(Aqua) - MOD(MYD)04-3K platforms between 2000-2020 at the studied area; (b) map of South America, highlighting the political limits of the studied region (magenta); (c) localization map micrometeorological tower in the Cláudia municipality, 50 km northeast of Sinop, Mato Grosso (white point). Changes in land use and land cover are also shown by TerraBrasilis.

Previous studies report the characteristics of this type of forest (Vourlitis et al., 2011), which typically have trees with lower height, biomass, and floristic diversity compared to humid tropical forests (Murphy and Lugo, 1986; Nogueira et al., 2008) due to their well-defined seasonal variation in precipitation. The forest is 423 m above sea level, in a transition where the vegetation consists of Savannah (Cerrado), transitional vegetation (Cerradão), and Amazonian forest (Vourlitis et al., 2011).

The areas of transitional forests (Amazon Forest-Cerrado) covered approximately 41% (362,538 km$^2$) of the State of Mato Grosso. Due to the advance of the agricultural frontier, 21% of these areas suffered drastic reductions. Part of these forest areas are found in protected areas and territories of indigenous communities (approximately 17%). The deciduous and semideciduous forests of the Cerrado biome initially covered 49,951 km$^2$ in the State of Mato Grosso. Deforested areas represented $\approx$ 41% of this total, with only 14% located in conservation units (Alencar A.; Nepstad D.; MacGrath D.; Moutinho, 2004). The

geographic positions of these forests are discontinuous due to climatic fluctuations that have occurred in the last 10,000 years (Prado and Gibbs, 1993). Trees at this location are typical of the semideciduous forest of the Amazon, with maximum canopy heights varying between 25-28 m. A comprehensive description of the species reported in the region was reported by Ackerly et al. (1989) and Lorenzi (2002). The soils are acidic with a pH measuring 4.2 and sandy (94% sand), well-drained quartzarenic

neosols, poor in nutrients, and with low organic matter (Vourlitis et al., 2001; Oliveira and Marquis, 2002), with a dry season that extends from May to September (Vourlitis et al., 2002). This area's 30-year average annual temperature is 24°C, with precipitation of approximately 2000 mm yr$^{-1}$ (Vourlitis et al., 2002). The Bolivian High (BH) and South Atlantic Convergence Zone (SACZ) are among the active atmospheric systems in northern Mato Grosso, while southern Mato Grosso is affected by extratropical systems, such as Frontal Systems (Reboita et al., 2012). The loss of leaves (deciduousness) during the dry season

(July-September) is sensitive to water availability and temperatures (maximum and minimum) in the region. With the arrival of the rainy season (November-May), trees that have lost their leaves begin to sprout again and produce new leaves (Vourlitis et al., 2011).

## 2.2  Instrumentation and Data

### 2.2.1  Aerosol Measurements

This study used a long series of aerosol optical depth measurements – AOD (Aerosol Optical Depth) to assess the impact of atmospheric particles on the flux of solar radiation to the surface. Two types of remote sensors were used: the MODIS (Moderate Resolution Imaging Spectroradiometer) orbital sensor, available on board the AQUA and TERRA satellites, products MOD04-3K and MYD04-3K (Remer et al., 2013), and an AERONET (Aerosol Robotic Network) solar photometer, used as a standard measure of optical properties of atmospheric aerosols at the surface, between June1999-March2017 (Holben et al.,

1998). All remote aerosol information required for this study was operated and maintained by NASA (National Aeronautics and Space Administration).

The TERRA /AQUA satellites have a heliosynchronous polar orbit, with a Local Time (LT) of passage over the study areas around 10h30min and 13h30min. These space platforms cover the Earth's surface every 1-2 days with radiance measurements in 36 spectral bands. The MOD/MYD043K aerosol products also feature the most current collection of data available from

NASA, currently at 3 Km spatial resolution for AOD and other aerosol optical properties (Levy et al., 2013; Remer et al., 2013). Filters to exclude contamination of data by clouds are also applied during estimation processing. The AOD series from these satellites has 20 years of data on continents and oceans and is widely available on the open access platform of the Atmospheric Files Distribution System – Level 1, located at the Distributed Active Files Center (LAADS-DAAC) from Goddard Space Flight Center – GSFC, in Greenbelt, Maryland (USA). In this work, satellite AOD spatializations were used

to obtain regional information on the nature or type of aerosol acting over the study area between 2000-2020 (Fig. S4). More detailed information about the MODIS sensor, such as spectral models, validation, and operating period of the aforementioned products can be found in Remer et al. (2005, 2013).

A long series of AOD measurements ($> 20$ years of data) are available for the city of Alta Floresta in northern Mato Grosso through CIMEL Electronique solar photometers, maintained and operated by NASA (GSFC), through the AERONET network (1999-2017). This photometer network is intended for the monitoring and characterization of aerosol particles in various regions of the world. These sensors represent the standard measure of AOD and are widely used in the validation of satellite AOD estimates. The system operates solar radiation measurements and rotational interference filters to extract optical properties from aerosols in various spectral bands, between 340-1020 nm (Schafer et al., 2002b, a; Procopio et al., 2004; Schafer et al., 2008). This makes it possible to evaluate the direct influence of atmospheric particles in real time on regions highly affected by fires, such as the region of the arc of deforestation. In this work, AOD was used at wavelengths of 500 nm (AERONET) and 550 nm (MODIS). Both satellite and photometer data cover the entire period of micrometeorological and flux data, described in the next section. In the Alta Floresta, the AERONET system also has individual sensors and long-term measurements of incident shortwave solar radiation ($SW_i(t)$), as described in Table 1.

### 2.2.2 Micrometeorological Measurements

The $CO_2$ flux data set available for this research were widely used and cited by previous studies. Information regarding the systems installed in the micrometeorological tower is directly available in Vourlitis et al. (2011). An automatic weather station (ASW) to monitor the weather in the Cláudia municipality was used between Jun2005 and Jul2008. The implanted tower follows the standard of the micrometeorological measurement tower system of the Programa LBA (Nagy et al., 2016; Artaxo et al., 2022). In this research, the deployed tower consists of a pyranometer, thermometer, psychrometer, anemometer, pluviograph, and a turbulent vortex system (*eddy covariance*). Herein, these measures were used to represent the biophysical factors that affect the photosynthetic rates of forests. Micrometeorological data were measured every 30-60 s and stored by data-logger systems (CR5000) and (CR-10X), both Campbell Scientific, Inc., from which hourly averages were calculated (Vourlitis et al., 2011). The micrometeorological data set used in this work is the same used in the study prepared by Vourlitis et al. (2011), whose data are previously validated. Technical details such as precision, accuracy, and calibration can be found in Vourlitis et al. (2011); Moreira et al. (2017). All direct measurements used are listed in Table 1.

### 2.2.3 Measures of flux and concentration of $CO_2$

The eddy covariance system has been widely used to measure the net $CO_2$ flux by the ecosystem. This system performs measurements by correlation of turbulent vortices from a sonic anemometer and an infrared gas chamber (Infrared Gas Analyzer, IRGA), from which flux measurements of $CO_2$ (Carbon), water vapor ($H_2O$) and energy (sensible heat – H and latent heat – LE) are determined at high frequency, usually 10Hz. The data generated and recorded by the *eddy* system, deployed in flux towers, is normally adjusted by compilation software such as Alteddy 3.90 (Alterra, WUR, Netherlands). The carbon flux data from these micrometeorological towers are presented, using the classical sign convention in atmospheric science (negative flux indicates net ecosystem $CO_2$ uptake).

**Table 1.** List of measured variables and instrumentation used in the micrometeorological tower (at Cláudia Municipality) and AERONET station, in Alta Floresta. The *flags* [1], [2] and [3] indicate the instrumentation used in the flux tower, AERONET system and AQUA space platforms (TERRA), respectively.

| Data set | Instrumentation | | Attributes | | |
|---|---|---|---|---|---|
| Measurements | Sensors [sites] | Models, Manuf. | Units | Symbols | Height |
| Inc. Solar Radiation | Pyranometer [1] | LI-200SB, LI-COR | $Wm^{-2}$ | $SW_i(t)$ | 40.0 m |
| Photosyn. Active Rad. | Pyranometer [1] | LI-190SB, LI-COR | $Wm^{-2}$ | $PAR_i$ | 41.5 m |
| Atmospheric Pressure | Barometer [1] | PTB101B, VSLA | hPa | $P_{air}$ | 42.5 m |
| Air Temperature | Thermohygrometer [1] | CS215, RMS | °C | $T_{air}$ | 41.5 m |
| Relative Humidity | Thermohygrometer [1] | HMP-35, VSLA | % | $RH_{air}$ | 41.5 m |
| Precipitation | Pluviometer [1] | GAUGE, MANUAL | mm | PRP | 40.5 m |
| Wind Speed | Sonic Anemometer [1] | CSAT-3, CSCI | $ms^{-1}$ | $US_s$ | 42.0 m |
| Wind Direction | Sonic Anemometer [1] | CSAT-3, CSCI | deg | $US_d$ | 42.0 m |
| $CO_2$ Flux | Eddy system [1] | LI-COR | $\mu mol\ m^{-2}s^{-1}$ | $FCO_2$ | 42.0 m |
| $CO_2$ Vertical Profile | IRGA [1] | LI-820, LI-COR | ppm | $[CO_2]$ | 1-28 m |
| Inc. Solar Radiation | Pyranometer [2] | CM21, K&Z | $Wm^{-2}$ | $SW_{ia}$ | – |
| Aerosol Optical Depth | Photometer [2] | CIMEL | - | $AOD_a$ | – |
| Aerosol Optical Depth | Modis-Terra [3] | MOD043K | - | $AOD_m$ | – |
| Aerosol Optical Depth | Modis-Aqua [3] | MYD043K | - | $AOD_m$ | – |

## 2.3   Methods for calculating NEE and radiative effects of aerosols

### 2.3.1   Method to determine the net exchange of $CO_2$ in the ecosystem

The $NEE$ is obtained from the eddy-covariance system. The eddy system provides $CO_2$ flux measurements at 10 Hz from a sonic anemometer (CSAT-3, Campbell Scientific, Inc., Logan, UT) integrated with an open-path gas analyzer (LI-7500, LI-COR Inc., Lincoln, NE). For $NEE$ calculation, the storage term $S[CO_2]_p$ is obtained according to Aubinet et al. (2012) and Araújo et al. (2010). For $S[CO_2]_p$ term calculation, we considered continuous measures of the $CO_2$ concentration vertically arranged between the ground and the top of the tower (Vourlitis et al., 2011). Under these conditions, the $NEE$ of $CO_2$ is approximated by Equation 1:

$$NEE \approx FCO_2 + S[CO_2]_p \tag{1}$$

where $FCO_2$ is called "$CO_2$ turbulent flux", calculated by the *eddy* system, above the treetops (Grace et al., 1996; Burba, 2013); $S[CO_2]_p$ is the vertical profile of the concentration of $CO_2$ or storage term (storage), considered a non-turbulent term measured at discrete levels $z$, at thicknesses $\Delta z_i$, from near the ground surface to the point of measurement of covariance of turbulent vortices in the tower (Finnigan, 2006; Araújo et al., 2010; Montagnani et al., 2018). In this work, the vertical profile $S[CO_2]_p$ was stratified into 5 reference levels (1, 4, 12, 20, and 28 m) (Vourlitis et al., 2011). Typical diurnal conditions consist of vector winds with speeds of 2.0 $ms^{-1}$ and $u^\star \geq 0.20$ m s$^{-1}$ and predominant SSW and SE directions. Approximately 72% of the

accumulated flux originates within 1 km and the representativeness of the measured $CO_2$ flux (footprint) is approximately 520

175 m (upstream of the tower), following the model proposed by Schuepp et al. (1990). The concentrations $[CO_2]$ were calculated

following Aubinet et al. (2001) and Araújo et al. (2010), as reported by Vourlitis et al. (2011).

$$S[CO_2]_p = \frac{P_{air}}{RT_{air}} \int_0^z \frac{\partial [CO_2]}{\partial t} \, dz \tag{2}$$

Where: $P_{air}$ is the atmospheric pressure ($Nm^{-2}$), R is the molar constant of the gas ($Nm \ mol^{-1} \ K^{-1}$) and $T_{air}$ the air temperature in Kelvin (K).

We also calculated $GPP$ from the $NEE$ data and estimates of ecosystem respiration ($R_{eco}$) obtained from the nighttime

$NEE$ (see Supplemental information), however, relationships between the atmospheric optical properties and $NEE$ were

qualitative similar to those using estimated $GPP$ (Fig. 9). Given the potential errors associated in estimating $GPP$ and $R_{eco}$

from the $NEE$ data (Reichstein et al., 2005), we decided to use the measured values of $NEE$ in our analysis of the impact of

atmospheric aerosols on land-atmosphere $CO_2$ exchange.

### 2.3.2 Method to determine the solar irradiance of clear sky

The term clear sky was used here to designate the minimal influence of clouds and aerosols on the solar radiation measured by

the pyranometer. To estimate the amounts of direct solar radiation to the surface under minimally overcast sky conditions, the

measurements $SW_{ia}$ of the AERONET 2.0 system observed under clear-sky (*cloudless*) conditions were used, that is, AOD $\leq$

0.10 (Artaxo et al., 2022), lacking fire plumes. Under these conditions, we get Equation 3; a polynomial fit of order 4, here

considered representative of the entire solar spectrum (Meyers and Dale, 1983). The model $S_0(t)$ obtained was used to derive

the clear-sky instants at the surface (Fig. S7) between 07-17h (LT), according to the formulation below:

$$SW_{ia}\{AOD \leq 0.10\} \approx S_0(t) = at^4 + bt^3 + ct^2 + dt + e \tag{3}$$

Where $S_0(t)$ is the clear-sky solar irradiance as a function of time, in $Wm^{-2}$. The parameters $(a, b, c, d, e)$ are the coefficients

of the polynomial curve and $t$, the time, in local hours (LT). Figure 2 shows the mean diurnal cycle of the $SW_{ia}$ obtained

from long-term aerosol measurements by the AERONET system under different pollution conditions. The plot illustrates

the sensitivity of the method applied to determine the expected irradiance levels on the canopy forest ($S_0(t)$) under varied

atmospheric aerosol loads (AOD), C2, C4, and C6 curves. Markers C1, C3, and C5 represent averaged observations between

07:00-17:00 used to fit C2, C4, and C6 curves.

Our methods consider the cloud-screened database AERONET (Figure 2). Using the long series of measurements of $AOD_a$,

it was possible to obtain different curves $S_0(t)$ for each month of the year, considering the seasonal variations of the $SW_{ia}$

given in Equation 3. Figure S7 shows the seasonal variation of the $S_0(t)$ diurnal cycle throughout the year. The coefficients of

the fit curves are listed in Table S1. To assess the consistency of the $S_0(t)$ model, obtained by $SW_{ia}$ AERONET data set, we

compared the outputs calculated by Equation 3 with the clear-skies solar irradiance model available by the Meteoexplration

(SolarCalculator). The Solar-Calculator is a free system used to compute the clear sky solar irradiance, managed by Meteo

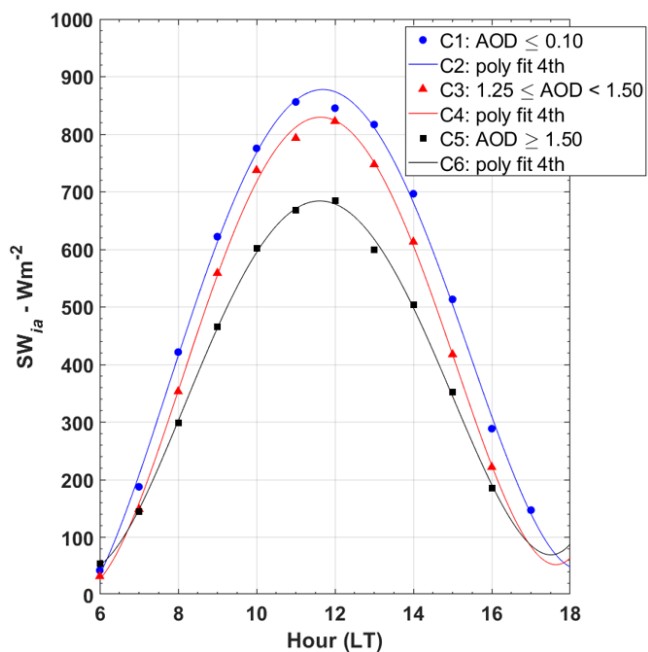

**Figure 2.** Incident solar irradiance under different sky conditions in Alta Floresta (June1997-March2017): clear-sky (C1 points and C2 curve, AOD ≤ 0.10) and polluted skies (C3 points and C4 curve, 1.25≤AOD≤1.50 C5 points and C6 curve, AOD>1.50)

Exploration company. The solar irradiance is calculated according to Bird and Hulstrom (1981), updated by Corripio (2003). The hyperlinked for Solar-Calculator is listed in Table S5. We also stated our algorithm is not accurate enough to separate shallow clouds of Biomass Burning Organic Aerosols plumes (BBOA). However, as the optical properties of deep clouds differ from aerosol (Machado et al., 2020), it is possible to distinguish them because the shading standard is notably different (Doughty et al., 2010). For example, deeper clouds often yield $PAR(D)_F \sim 1$ (unit) meanwhile high loads of BBOA are usually lower than < 1 (unit). In these conditions, the parameter $f$ can be used as a sensible indicator of aerosol presence/entrance (Gu et al., 2003; Jing et al., 2010) but not to detect shallow clouds (translucent). In practice, the uncertainties for the radiative effects of aerosols on NEE are greater when $PAR(D)_F$ is near the unit. Thus, our algorithm cannot state the complete absence of clouds but is a crucial qualitative indicator of aerosols in the atmosphere when sunphotometers are not promptly available.

### 2.3.3    Determination of relative irradiance

In practical terms, the relative irradiance $f$ expresses the relationship between incident solar radiation and that observed at the surface under a clear-sky (AOD ≤ 0.10) and cloudless ($f \geq 1.0$). It is a parameter indicating the presence of pollution plumes with aerosols that scatter solar radiation and clouds, generally used in areas without direct instrumentation of clouds. This parameter has been considered a key indicator in the detection of clouds and plumes of pollution from fires in the Amazon Gu

et al. (2001); Oliveira et al. (2007); Cirino et al. (2014). To this end, the observed amounts of $SW_i(t)$ on the forest canopy are

normalized by the irradiance $S_0(t)$, thus determining the quotient $f$ (a dimensionless parameter), according to Equation 4. It is also highlighted that values of $f$ can assume values as large as 1.2-1.3 ($f \sim$20-30%), typically due to the so-called "cloud gap effect" (Duchon and O'Malley, 1999; Gu et al., 2001). There is still no consensus on values in the literature about it. This term denotes the cloud-induced increase in surface irradiance. In general, there are multiple scatterings of solar radiation by the clouds around the study area, but still outside the Pyranometer's viewing angle. It will be seen in more detail in a few sections

ahead.

$$f = \frac{SW_i(t)}{S_0(t)\{AOD_a \leq 0.10, \text{cloudless}\}} \tag{4}$$

Where $SW_i(t)$ is the total incident solar irradiance measured by the pyranometer ($Wm^{-2}$) under any atmosphere and $S_0(t)$ is the clear sky solar irradiance ($Wm^{-2}$) on a flat surface perpendicular to the sun's rays, without the attenuating effects of the atmosphere (clouds and burned) for a given time and place, ie $AOD_a \leq 0.10$ (*cloudless*). Values close to zero represent cloudy

and/or smoky-sky conditions, and values close to unity represent clear-sky conditions (Gu et al., 1999; Oliveira et al., 2007; Jing et al., 2010; Cirino et al., 2014; Gao et al., 2021).

Here, we used $f$ as a basis for comparison to detect the joint presence of clouds and aerosols from fires over the study area since the experimental site does not have instrumentation for direct observation of cloud cover. Obtaining this parameter is extremely important because when using clear-sky solar radiation as a base, solar radiation measured under overcast skies

becomes a new metric for observing cloudiness. This variable will be compared with the $NEE$ to assess the photosynthetic responses of the ecosystem to variations in the external environment.

### 2.3.4    Determining the clarity index

To determine the brightness index $kt$ the extraterrestrial solar irradiance $S_{ext}$ was first calculated depending only on orbital parameters. The index $kt$ is a coefficient of proportionality between the measurements of direct solar radiation to the surface

and $S_{ext}$ and expresses the direct solar radiation transmitted in the atmosphere (Gu et al., 1999; Cirino et al., 2014). In a first approximation, $kt$ indicates the transmissivity; the degree of transparency of the atmosphere to solar radiation at a given time and place, while $f$ is a parameter of comparison more sensitive to the presence of radiation-scattering aerosols and clouds. Here, $kt$ and $SZA$ were used as predictors of the diffuse component of radiation (Gu et al., 1999; Cirino et al., 2014). For the calculation of the irradiance $S_{ext}$ some parameters and variables are also needed such as the solar constant of the Earth

($S_{ext}^t$), the latitude of the location ($\varphi$), solar declination ($\delta$), hour angle ($h$) and mean square distance between the Earth and the Sun (Gates, 1980). The determination of $S_{ext}$ takes into account the angle of incidence of the solar rays and, therefore, the variations in the amounts of solar radiation at the surface, modulated by the $SZA$. Under these conditions, $kt$ can be expressed according to Equation 5:

$$kt = \frac{SW_i(t)}{S_{ext}} \tag{5}$$

Where $SW_i(t)$ is the short wave radiation (Wm$^{-2}$) measured by the pyranometer (Table 1) and $S_{ext}$ the extraterrestrial solar irradiance (Wm$^{-2}$) estimated on a perpendicular surface to the sun's rays, without the attenuating effects of the atmosphere for a given time and place, expressed according to Equation 6:

$$S_{ext} = S_{ext}^{T} \left( \frac{\bar{D}}{D} \right)^{2} \times cos(z) \tag{6}$$

In this equation $S_{ext}^{t}$ is the Earth's solar constant ($\approx 1367$ Wm$^{-2}$), $\bar{D}$ is the average earth-sun distance ($\sim 1.49 \times 10^{6}$ km), D is
the earth-sun distance on a given Julian day, and cos (z) in the cosine of the solar zenith angle ($SZA$), calculated as proposed by Bai et al. (2012). This calculated index was used to establish the diffuse solar radiation, as described in detail in the next section.

### 2.3.5    Determination of diffuse PAR radiation

To determine the Diffuse component of the total PAR ($PAR(D)$), we adopted the procedures of Spitters et al. (1986) and
Reindl et al. (1990), widely used in the literature when there are no direct measurements of radiation $PAR(D)$ (Gu et al., 1999; Jing et al., 2010; Zhang et al., 2010; Bai et al., 2012). The detailed calculation can be found in the one performed by Gu et al. (1999). The estimate is performed by deriving the diffuse PAR radiation according to the formulation below (Spitters, 1986).

$$PAR(D) = \left[ \frac{1 + 0.3 \left( 1 - q^2 \right) q}{1 + \left( 1 - q^2 \right) \cos^2 \left( 90 - z \right) \cos^3 \left( z \right)} \right] \times PAR_i \tag{7}$$

Where $PAR(D)$ is the incidence of the diffuse (total) PAR ($\mu$mol photon m$^{-2}$s$^{-1}$), in the near-infrared range, in a horizontal plane to the Earth's surface, while $q$ is a coefficient of proportionality used to denote the ratio of the total diffuse radiation to a given amount of irradiance ($SW_i$) at the surface given the sky conditions (Wm$^{-2}$). The parameter $q$ is expressed considering ranges of variation for the index $kt$ (Gu et al., 1999). To express the diffuse fraction of PAR radiation ($PAR(D)_F$) we use the relationship between $PAR(D)$ and $PAR_i$ (Spitters et al., 1986). In the absence of direct measurements of diffuse solar
radiation, the procedures reported by these authors are still widely used (Jing et al., 2010; Cirino et al., 2014; Moreira et al., 2017).

### 2.3.6    Determining the efficiency of light use

Another important parameter in this study is the light use efficiency ($LUE$), which expresses the efficiency of light use in photosynthetic processes by the canopy and is defined as the ratio between $NEE$ and $PAR_i$. Several other procedures have
been used to approximate the $LUE$; some use the coefficient of proportionality between the $NEE$ and the $PAR(D)$ (Moreira et al., 2017) radiation, and others use temperature measurement directly on the leaf of the trees (LI-COR) to capture the photosynthetic response as a function of the variation in light intensity (Doughty et al., 2010). Canopy radiative transfer codes with validated physical parametrization for different leaf types are also used (Mercado et al., 2009). Here, for practical reasons,

we used the procedures applied by Jing et al. (2010) and Cirino et al. (2014), according to Equation 8, where $LUE$ is given in percentage values.

$$LUE \cong \frac{NEE}{PAR_i} \tag{8}$$

We also performed the same procedure with $GPP$, but as mentioned above our results with $GPP$ were qualitatively similar to those obtained using $NEE$. Since $NEE$ was measured directly with only assess $LUE$ calculated from $NEE$.

### 2.3.7 Determining leaf canopy temperature

We used the parameterization proposed from (Tribuzy, 2005) to estimate leaf canopy temperature (LCT) obtained from field experiments carried out in central Amazonia, 60-70 km NW from the center of Manaus-AM. Thermocouples temperature measurements on leaves provided a significant statistical relationship between $PAR_i$ and $RH_{air}$ during both dry (July-August/2003) and wet seasons (December 2003 to February 2004). The final equation obtained is expressed as a function of relative air humidity ($RH_{air}$) and radiation ($PAR_i$), valid for dry and wet seasons (Equation 9).

$$LC_T = [(2.48 \cdot 10^{-6}(\mathrm{RH}_{air})^2 - 1.82 \cdot 10^{-4}(\mathrm{RH}_{air}) - 1.83 \cdot 10^{-6}(\mathrm{PAR}_i) + 0.0363)]^{-1} \tag{9}$$

Where $LC_T$ is leaf temperature of canopy (°C), $PAR_i$ and $RU_{air}$ are photosynthetically active radiation ($\mu$mol m$^{-2}$s$^{-1}$) and relative humidity (%), respectively. Due to uncertainties and limitations underlying Equation 9, we also used an alternative method based on the Stefan-Boltzmann equation (Equation S1), following Doughty et al. (2010) and Cirino et al. (2014) (Figs. S9 and S10), results discussed in subsection 3.7.

### 2.3.8 Determination of clear sky NEE

The $NEE$ observed on clear days (AOD $< 0.1$ and clear) was also used as a basis for comparing days with high aerosol loading. The Fig. 3 illustrates the behavior of the $NEE$ under clear sky conditions ($f \approx 1.0$) between 07-17h (LT). The obtained polynomial fits are used to determine the $NEE_0(sza)$ as a function of $SZA$ variations for each month of the year between Jun./2005 and Jul./2008 (Figure 3). We listed the curve coefficients in Table S2. The estimated curves and their goodness of fit are consistent with the behavior observed in previous studies (Gu et al., 1999; Cirino et al., 2014). We have used Equation 10 to estimate the $NEE_0(sza)$ throughout the year, considering the seasonal changes of biophysical factors such as solar radiation, deciduousness, water, and heat stress, that may add time-dependent noise to the fitted model. Figure S8 shows seasonal changes on the $NEE_0(sza)$ (hourly mean cycle).

$$NEE_0(sza) = p_1 SZA^2 + p_2 SZA + p_3 \tag{10}$$

Where $NEE_0(sza)$ is the $NEE$ typically found on clear sky days ($\mu$mol m$^{-2}$s$^{-1}$). The parameters $p_1$, $p_2$, and $p_3$ are the coefficients of the polynomial curve and equal 0.0038, $-0.99$, and $-12$, respectively.

Like $f$, $\%NEE$ was used here as a basis for comparison for the maximum negative values observed during the study period, assuming the absence of water stress and nutrient deficiency (Gu et al., 1999; Oliveira et al., 2007; Doughty et al., 2010; Cirino et al., 2014).

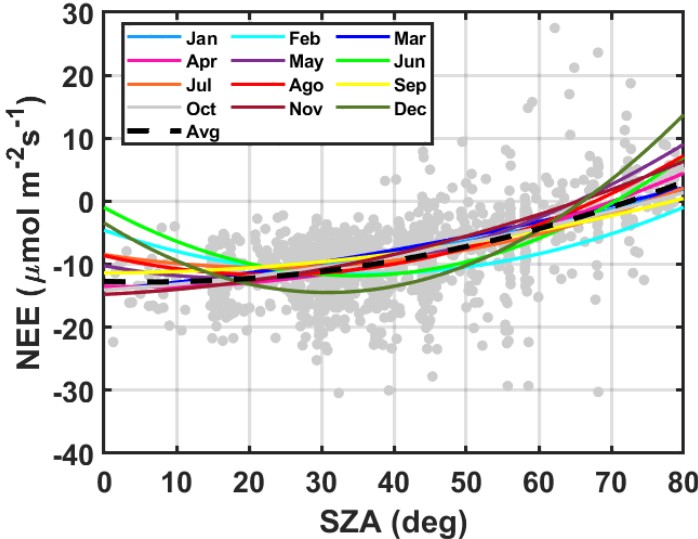

**Figure 3.** Show the $NEE$ monthly changes as a $SZA$ function to clear-sky conditions ($f \sim 1.0$) from 07 to 17h (LT), between Jun2005-Jul2008. Fitted curves coefficients $NEE_0(sza)$ were listed in Table S2. The black dot line is the annual average curve $NEE_0(sza)$.

Changes in observed $NEE$ versus $NEE$ under clear sky conditions were used to determine the percentage effect of aerosols on $NEE$. The $\%NEE$ was calculated by the following relationship (Bai et al., 2012; Gu et al., 1999; Oliveira et al., 2007):

$$\%NEE = \left( \frac{NEE(sza) - NEE_0(sza)}{NEE_0(sza)} \right) \times 100 \tag{11}$$

To eliminate solar elevation angle interference in the analysis of changes in $\%NEE$ versus $f$, we grouped the data into $SZA$ ranges of 20-25°.This interval was small enough to minimize the effects of solar uplift during the day and to represent changes in $NEE$ as a function of $f$ in response to aerosols and/or clouds alone. This interval also ensured sufficient sample size for statistical analyses. $SZA$ intervals smaller than 15° significantly reduced the sample size, making it impossible to develop a robust statistical analysis (Gu et al., 1999). Values above 50 or around 0 (solar angles very close to the horizontal and vertical plane, respectively) were generally very contaminated by clouds (Gu et al., 1999; Cirino et al., 2014).

## 2.4 Data analysis procedures

Computational routines were developed for compilation, certification, organization, and analysis of the variables presented in Table 1. We performed fitting curves and mathematical or statistical calculations with the packages available in MATLAB (2013). For data quality control, non-physical values outside acceptable levels were excluded from the database, totaling a loss of 3% of the total set of valid measurements (approximately 3,600 sampled points). We exclude unexpected maximum and minimum values for the region, e.g., values below and above 20-40°C, 40-95%, -40-40 ($\mu$mol m$^{-2}$s$^{-1}$), 0-1000 (Wm$^{-2}$) and

**Table 2.** List of indirect (calculated) variables, symbols, and measurement units of derived quantities, according to the cited body of literature.

| Indirect Measures | Symbols | Units | Literature |
|---|---|---|---|
| $CO_2$ Net Exchange | $NEE$ | $\mu$mol m$^{-2}$s$^{-1}$ | Vourlitis et al. (2011) |
| Gross Primary Productivity | $GPP$ | $\mu$mol m$^{-2}$s$^{-1}$ | Wutzler et al. (2018) |
| Ecosystem Respiration | $R_{eco}$ | $\mu$mol m$^{-2}$s$^{-1}$ | Wutzler et al. (2018) |
| Vapour Pressure Deficit | $VPD$ | hPa | Vourlitis et al. (2011) |
| Clear Sky Solar Irradiance | $S_0(t)$ | Wm$^{-2}$ | (Author) |
| Solar Zenith Angle | $SZA$ | Degrees | Bai et al. (2012) |
| Relative Irradiance | $f$ | - | Cirino et al. (2014) |
| Clarity Index | $kt$ | - | Gu et al. (1999) |
| Extraterrestrial Solar Irradiance | $S_{ext}$ | Wm$^{-2}$ | Gu et al. (1999) |
| Diffuse PAR Radiation | $PAR(D)$ | $\mu$mol phot. m$^{-2}$s$^{-1}$ | Gu et al. (1999) |
| Diffuse PAR Fraction | $PAR(D)_F$ | - | Gu et al. (1999) |
| Efficiency of Light Use | $LUE$ | - | Jing et al. (2010) |
| Leaf Canopy Temperature | $LC_T$ | °C | Tribuzy (2005) |
| Clear Sky NEE Exchange | $NEE_0(sza)$ | $\mu$mol m$^{-2}$s$^{-1}$ | (Author) |
| Relative NEE Exchange | $\%NEE$ | % | (Author) |

0-3000 ($\mu$mol m$^{-2}$s$^{-1}$) for T$_{air}$, RH$_{air}$, FCO$_2$, SW$_i(t)$, and PAR$_i$, respectively. Data analysis consists of three fundamental steps: (1) variation of solar radiation with optical depth AOD$_a$ analyzed as a function of irradiance $f$; (2) effects of aerosols and clouds on the net exchange of $CO_2$ at the forest-atmosphere interface and, finally, (3) quantification of photosynthetic performance as a function of pollution loads, and analysis of how pollution loads affected biological critical or optimal values for environmental factors such as T$_{air}$, $LC_T$ and $VPD$ (Vapour-Pressure Deficit). Photosynthetic performance, in all cases, is

analyzed as a function of $NEE$. In the end, the net percentage variation of the photosynthetic activity of the forest ($\%NEE$) is evaluated as a function of the irradiance $f$. Non-linear regression was used to determine functional relationships between $NEE$ and other radiation variables. The relationships found are evaluated from the Poisson correlation and tabulated in terms of basic descriptive statistical parameters such as coefficient of determination (R$^2$) and significance level (P$_{value}$) with confidence intervals of 95%. Basic descriptive statistics are also applied to the data to obtain mean values, medians, percentiles, and

standard deviations for the measured and estimated variables. Table 2 lists indirect variables, calculated from the dataset listed in Table 1.

## 3 Results and Discussions

### 3.1 Average daily cycle of net exchange of $CO_2$

The average daily pattern of $NEE$ observed in 2005-2008 (Fig. 4) follows the typical pattern of tropical forests (Gu et al.,
1999; Niyogi et al., 2004; von Randow et al., 2004; Araújo et al., 2010; Vourlitis et al., 2011). Figure 4 shows maximum
negative fluxes average $-13.7 \pm 6.2$ $\mu$mol m$^{-2}$s$^{-1}$ around 10-11h (LT), and the maximum positive fluxes average $+6.8 \pm 5.8$
$\mu$mol m$^{-2}$s$^{-1}$ during the night period between 19h and 05h (LT). We observed a slight difference in the pattern of the daily
cycle of the $NEE$ between the wet and dry seasons (Fig. 4), with shift (an advance) in the peak absorption of $CO_2$ from the
wet-to-dry season, from about 12h (LT) to 10h (LT), respectively (Fig. 4). Our estimates of $CO_2$ absorption were about 10-15%
lower (i.e, less negative) during both seasons ($< 0.6$ $\mu$mol m$^{-2}$s$^{-1}$) when compared to Vourlitis et al. (2011). We hypothesize
seasonal variations in water availability, nutrients, radiation, temperature, $VPD$, and pollution are counterbalanced throughout
the year, producing an average seasonal behavior without significant differences in $NEE$.

Furthermore, different approaches in both studies can also explain the differences, i.e., analyses performed on different
time scales. For example, (Vourlitis et al., 2011) reported average $NEE$ of $CO_2$ values from daily and monthly time series.
Similar monthly variations, with more negative magnitudes during the day in the rainy months ($-9.0$ $\mu$mol m$^{-2}$s$^{-1}$ between
November-February) and less negative during the light hours in the dry months ($-7.7$ $\mu$mol m$^{-2}$s$^{-1}$ between May-August)
were observed. The general balance of NEE revealed net carbon uptake of $-0.12$ $\mu$mol m$^{-2}$s$^{-1}$ and $-0.18$ $\mu$mol m$^{-2}$s$^{-1}$
during the wet and dry seasons, respectively. The maximum rates of photosynthesis and leaf canopy respiration were observed
in October-November, which are the first months of the rainy season.

### 3.2 The influence of aerosols on short wave solar radiation

The impact of aerosol particles by fires on the SW$_i$ flux is evaluated as a function of $f$, AOD$_a$, $SZA$, $PAR(D)_F$ and PAR$_i$.
Fig. 5 (a) shows the behavior of the relative irradiance $f$ for different levels of AOD$_a$ pollution, in the $SZA$ ranges between 20-
50°. A close and statistically significant relationship between $f$ and AOD$_a$ is observed with p-value $< 0.01$ and a coefficient
of determination R$^2 \approx 0.92$ (Table 3). An approximately linear relationship is observed in which $f$ decreases by about 40-
60% when the AOD$_a$ varies from 0.10 to 5.0, and no statistically significant difference was observed between mornings and
afternoons. There is only a slight increase of $\approx$ 5-20% (on average) in the value of $f$ between late mornings and afternoons,
attributed here to the multiple scattering of solar radiation due to the formation of clouds near the tower (Gu et al., 2001). For
$SZA$ angles between 20 and 50°, there is a strong reduction in SW$_i$ ($225 \pm 50$ Wm$^{-2}$) associated mainly with the increase in
the concentration of aerosols emitted by local fires or transported regionally during the burning season. Oliveira et al. (2007)
and Cirino et al. (2014) reported results about 2-3 times lower for 20-30% reductions in $f$ and AOD increase from 0.1 to 0.8,
in FLONA-Tapajós (Santarém-PA) and central Amazon (K4), in Manaus-AM.

Figure 5 (b) shows the fraction of diffuse radiation calculated as a function of AOD$_a$, with a close statistical relationship
observed (R$^2$ = 0.98 and 0.96) for the morning and afternoon hours (Table 3). Due to the reduction in the instantaneous fluxes
of SW$_i$ an increase of about up to 85% in diffuse radiation is observed when the AOD$_a$ increases from 0.10 to 5.0. These

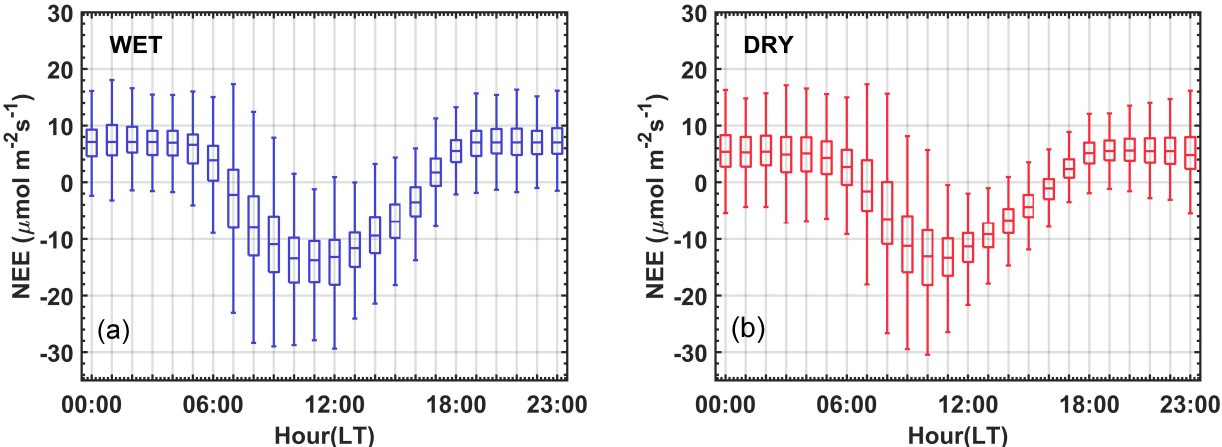

**Figure 4.** $NEE$ average hourly cycle between June/2005 and July/2008, during the rainy (a) and dry (b) seasons for the semideciduous forest at the Claudia municipality. No filters are applied. The $NEE$ is presented for any sky conditions during the year. We used the box plot to represent the distribution of $CO_2$ flux data. The vertical bars are the maximum and minimum values. The lower and upper limits of the boxes represent, respectively, the 25th and 75th percentiles, whereas the horizontal blue and red lines represent the median of the $CO_2$ flux data.

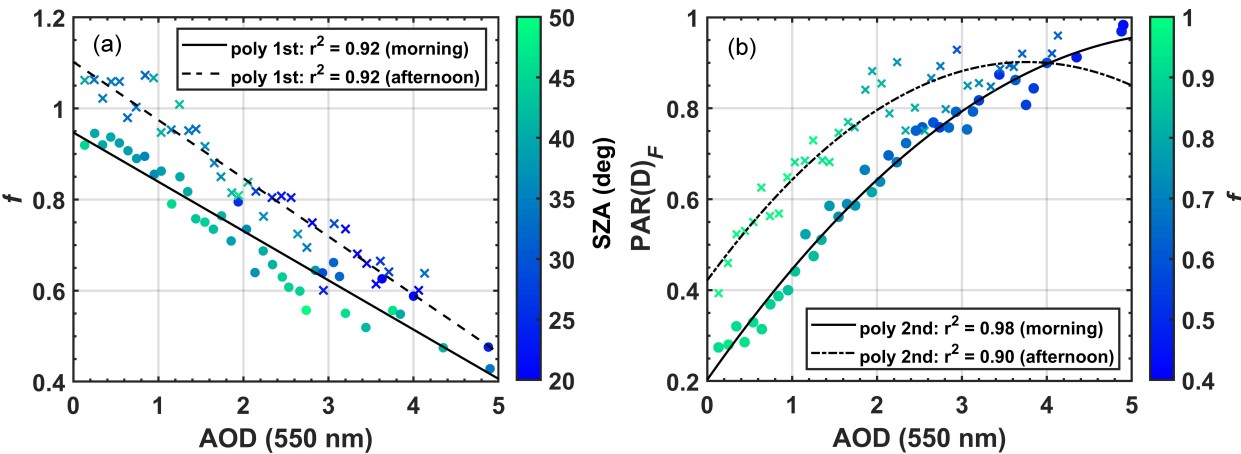

**Figure 5.** 3D-correlation between $f$ and $PAR(D)_F$ with increasing $AOD_a$ for different values $SZA$ (a) and irradiance $f$ (b) in semideciduous forest in the Cláudia municipality, 50 km northeast of Sinop-MT (2005-2008). Means are presented as clustered (bin) points, quantified, and determined in terms of standard deviation (STD) for each bin (STD shown in Table 3).

results are consistent with previous studies carried out in the Brazilian Amazon (Doughty et al., 2010; Cirino et al., 2014; Rap, 2015; Moreira et al., 2017; Malavelle et al., 2019; Bian et al., 2021) and around the world (Niyogi et al., 2004; Jing et al., 2010;

Rap, 2015; Rap et al., 2018) and proves to be particularly important due to the ability of $PAR(D)$ to penetrate more efficiently into the leaf canopy, and under certain conditions, increase ecosystem in carbon uptake.

**Table 3.** Polynomial adjustments (Fig. 5), coefficients and statistics for the morning and afternoon periods in the micrometeorological tower in Cláudia-MT (2005-2008). $R^2$ is the correlation coefficient, $\Delta SW_i$ is the incident shortwave radiation amount, and STD is the Standard Deviation.

| Settings | | Period | Coefficients | | | Statistics | |
|---|---|---|---|---|---|---|---|
| Polynomial Functions | | Local Hours | $a$ | $b$ | $c$ | $R^2$ | $\Delta SW_i$ (STD) |
| $f$ | poly fit 1st | 07-12h | $-0.11$ | 0.95 | $-$ | 0.92 | -200 ($\pm$ 50) |
| | $-$ | 12-17h | $-0.13$ | 1.10 | $-$ | 0.92 | -250 ($\pm$ 80) |
| $PAR(D)_F$ | poly fit 2nd | 07-12h | $-0.023$ | 0.27 | 0.20 | 0.98 | -97 ($\pm$ 30) |
| | $-$ | 12-17h | $-0.034$ | 0.25 | 0.42 | 0.90 | -118 ($\pm$ 42) |

### 3.3 The influence of aerosols on diffuse radiation

Figure 6 shows the behavior of PAR$_i$ and $PAR(D)$ as a function of $f$ and $SZA$. For reductions in $f$ of $\approx 40\%$ ($f$ ranging from 1.0 to 0.6) there were strong reductions in PAR$_i$ ($\sim 750 \ \mu$ mol m$^{-2}$s$^{-1}$) and a corresponding 55% increase in diffuse radiation $PAR(D)$ ($\sim 600 \ \mu$mol m$^{-2}$s$^{-1}$) between July-December. These numbers indicate a strong reduction in PAR$_i$ as pollution levels increase and change from clear sky conditions (AOD $\leq 0.10$, $f \sim 1.0$) to aerosol smoky sky conditions of fires (AOD $\gg 0.1$, $f \ll 1.0$). Figure 6a shows a decreases almost linearly between PAR$_i$ and $f$, meanwhile behavior $PAR(D)$

versus $f$ is non-linear (Figure 6b). The polynomial fits, coefficients, and inflection points are displayed in Table 4. $PAR(D)$ reach maximum values (779-1080 $\mu$mol m$^{-2}$s$^{-1}$) for $f$ between 0.63 and 0.66 (reductions of 37 %-34%) and ranges $SZA$ (20-40$^\circ$). The negative variations in $f$ also suggest high pollution load for fires at the site (AOD $\gg 0.10$) producing statistically significant reductions of up to 35% in the PAR radiation flux (Figs. 6a and 5a) and a 50% increase in $PAR(D)_F$ (Figs. 6b and Fig. 5b).

For SZA $< 40^\circ$, it is observed the higher variation rates PAR(D) $f^{-1}$, indicating the entrance/presence of plumes-pollution and clouds over the measurement tower's pyranometer. Table 4 shows a slight shift of the tipping points ($Cp$) towards smaller values of $f$ ($\sim$1.0-0.60), as well as an accentuated increase in PAR(D) (50%, $\sim$400-500 $\mu$mol phot. m$^{-2}$s$^{-1}$). These results are likely linked to greater optical thickness of the atmosphere at the beginning and end of the day and higher aerosol concentration (BBOA). Here, we raised two reasonable hypotheses: (1) lower PBL (Planetary Boundary Layer) favors higher

BBOA concentration over the tree canopy, usually between 06-09h LT (SZA $< 75^\circ$); (2) thicker PBL, provides deeper-clouds and higher shading on the canopy (Oliveira et al., 2020), beyond favors dispersion of fires Nepstad et al. (2014), intensifying BBOA concentration at local by advection or particle regional-transport (Figs. 1, S4-S6). For a given period of the year, under stable meteorological conditions, BBOA can explain changes in PAR(D), at least on an hour basis, especially between May and October, when evapotranspiration is more than rainfall (ET $>$ PRP) Vourlitis et al. (2002, 2011) and deeper cloud-cover

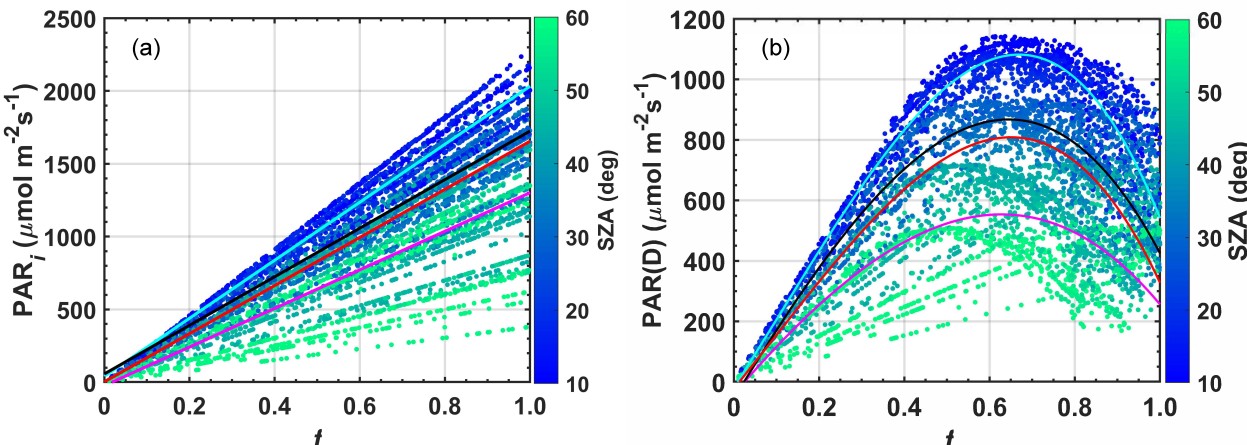

**Figure 6.** (a) 3D-correlation between $f$, PAR$_i$ (a) and $PAR(D)$ (b) for different $SZA$ values. The blue, black, magenta and red lines are the polynomial curves adjusted to the analyzed $SZA$ variation ranges, respectively equal to 0-20º, 20-40°, 40-60°, and 0-60º, in semideciduous forest in the Cláudia municipality, 50 km northeast of Sinop-MT (2005-2008).

fraction is lower often. In the other months of the year, clouds and aerosols mix, producing radiative effects inseparable, considering our instrumentation and dataset available at the local studied. The 50% increase in $PAR(D)$ can be mainly attributed

**Table 4.** Polynomial adjustments (Fig. 6), coefficients, and statistics for the morning and afternoon periods in the micrometeorological tower in Cláudia-MT (2005-2008). $\mathbf{C}p\,(x_v, y_v)$ is the critical point of the fit curve, where the derivative is equal to zero.

| Settings | Angles | Coefficients | | | | Statistic | |
|---|---|---|---|---|---|---|---|
| **Polynomial Functions** | **SZA** | ***a*** | ***b*** | ***c*** | ***d*** | $\mathbf{R}^2$ | $\mathbf{C}p\,(x_v, y_v)$ |
| PAR$_i$ | 0-20° | $+1.5 \times 10^3$ | $+56$ | | | 0.92 | |
| | 20-40° | $+2.0 \times 10^3$ | $+41$ | | | 0.86 | |
| poly 1st | 40-60° | $+1.7 \times 10^3$ | $+57$ | | | 0.64 | |
| | 0-60° | $+1.3 \times 10^3$ | $-23$ | | | 0.67 | |
| $PAR(D)$ | 0-20° | $-2.5 \times 10^3$ | $+8.4 \times 10^2$ | $+2.2 \times 10^3$ | $-19$ | 0.92 | (0.66, 1080) |
| | 20-40° | $-1.3 \times 10^3$ | $-5.6 \times 10^2$ | $+2.3 \times 10^3$ | $-56$ | 0.66 | (0.63, 846) |
| poly 3rd | 40-60° | $-6.4 \times 10^2$ | $-7.0 \times 10^2$ | $+1.6 \times 10^3$ | $-41$ | 0.42 | (0.61, 529) |
| | 0-60° | $-2.0 \times 10^3$ | $+5.8 \times 10^2$ | $+1.7 \times 10^3$ | $-22$ | 0.40 | (0.63, 779) |

to radiation-scattering particles (BBOA), especially during the dry season (Shilling et al., 2018; de Sá et al., 2019) indeed cloud-cover. On fire seasons, about 80% of BBOA is composed of fine particles PM$_{2.5}$ (Bian et al., 2021) from which 10% is BC (Black Carbon) and BCr (Brown Carbon), which both Single Scattering Albedo (SSA) and AOD can be affected. In

general, these particles have the potential to heat the atmosphere (absorption greater than reflection), producing values that may be above the optimal physiological thresholds of the ecosystem, influencing $CO_2$ absorption rates (maximum-negative $NEE$). It is also possible a mix of other kinds of particles from long-range transport with complex chemical properties, e.g., urban aerosols and African BBOA (de Sá et al., 2019; Holanda et al., 2023).

### 3.4 The indirect effect of aerosols on the use of light efficiency by the forest

There was a well-defined monthly variation of $AOD_a$, as shown in the previous sections. Since fires are the main cause of changes in the physical and chemical composition of the atmosphere throughout the year (Martin et al., 2010b, a; Artaxo et al., 2013, 2022), statistically significant reductions were found for the $SWi$ and $PAR_i$. This section mainly evaluates the optimal levels of $PAR_i$ as well as the effects of changes in the efficiency of solar radiation use by the forest ($LUE \approx NEE$/$PAR_i$). The analyses are performed as a function of $PAR(D)$ radiation, from which the maximum efficiency of light use for the studied semideciduous forest is determined. Under smoky sky conditions (AOD $\gg 0.10$), carbon assimilation gradually increases with increasing $PAR_i$ reaching maximum saturation around 1550 and 1870 $\mu$mol m$^{-2}$s$^{-1}$ in the range between 20-50° $SZA$, values for which the maximum $NEE$ (negative) is approximately $-23$ $\mu$ mol m$^{-2}$s$^{-1}$. Under clear sky conditions, considering the same $SZA$ range, the maximum negative $NEE$ is about around $-18$ $\mu$mol m$^{-2}$s$^{-1}$, which occurs with a $PAR_i$ of 2100-2300 $\mu$mol m$^{-2}$s$^{-1}$ (Fig. 7a). To complement this analysis, the $LUE$ flux normalized by $PAR(D)_F$ during days with high aerosol loading in the burning season (Fig. 7b). Under these conditions, the forest reaches maximum $NEE$ fluxes on smoky days and not under clear sky conditions. The results reveal that smaller amounts of energy are needed for the forest to reach maximum saturation on non-polluted days. The analyses presented in Fig. 7 confirm greater photosynthetic efficiency under smoky sky conditions for the studied semideciduous forest ecosystem, results compatible with field observations (Oliveira et al., 2007; Doughty et al., 2010; Cirino et al., 2014) and by numerical modeling in the Amazon (Rap, 2015; Moreira et al., 2017; Malavelle et al., 2019; Bian et al., 2021) and the world (Rap et al., 2018).

Due to the physicochemical nature of the BBOA and its intrinsic properties (Cirino et al., 2018; Adachi et al., 2020), the radiation $PAR(D)$ affects the $NEE$ and the functioning of several Amazon forest ecosystems (Rap, 2015; Rap et al., 2018; Bian et al., 2021), especially where tree species adapted to low light conditions occur, for example, in the leaf sub-canopy of Amazonian forests (Mercado et al., 2009).

Photosynthetic efficiency ($LUE$), closely linked to the canopy's ability to convert solar energy into biomass, is $\sim$ 1-2% for the studied forest, indicating loss or rejection of a large part of the solar energy available for photosynthesis. However, for high values of $PAR(D)_F$, close to 1.0, peaks of up to 3% in LUE are observed. In situations where the diffuse fraction total maximum values, the values of $AOD_a$ are on average $< 1.0$ and $f \ll 1.0$. These findings corroborate the previous analyses and reinforce the presence of radiation-scattering aerosols emitted by the fires over the studied area. It is noteworthy that changes in $PAR(D)_F$ (Fig. 7b) express proportional changes in $PAR(D)$.

Although there is great uncertainty (high standard deviation) in the behavior of $LUE$ with increasing radiation $PAR(D)$, there is a gradual, approximately linear increase in the values of $LUE$ in the range of radiation $PAR(D)_F$ between 0.20-1.0. This behavior is peculiar to tall vegetation with a generally leafy canopy of tropical forests, which are more sensitive to the

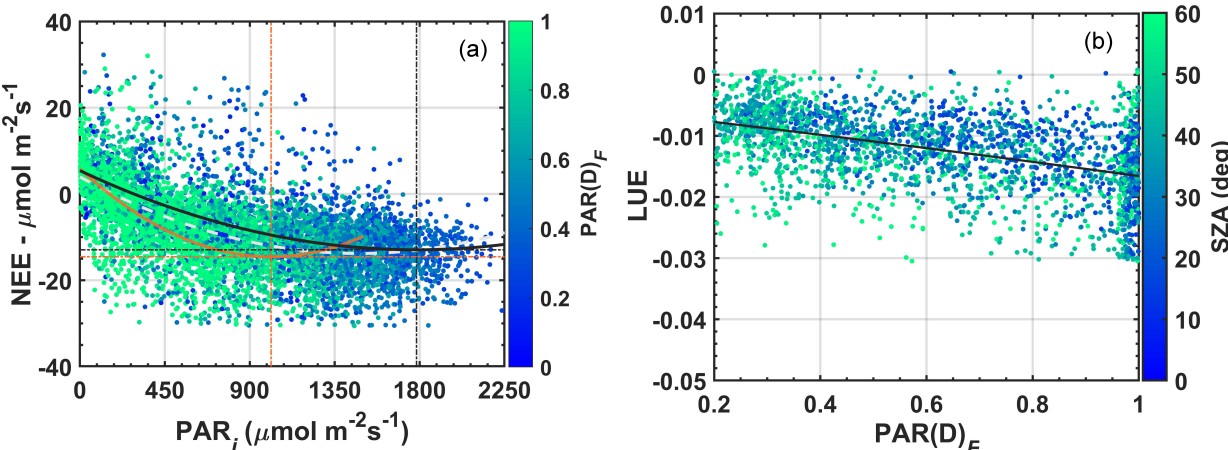

**Figure 7.** $NEE$ as a function of radiation, $PAR_i$ for measurements between 08h and 17h LT (a). In image (b) $LUE$ is a function of the fraction $PAR(D)_F$ ($R^2 = 0.21$, the value of $p < 0.001$) for an area of semideciduous forest located in the municipality of Cláudia- MT, 50 km north of Sinop, between Jun2005-Jul2008. The orange and black lines denote, respectively, observations $PAR(D)_F \geq 0.60$ and observations $f \sim 1.0$ (clear-sky conditions). The orange and black vertical lines indicate the global minima of the polynomial curves.

transfer of $PAR(D)$ radiation from the top canopy to the bole. In short-stature vegetation, as in the semiarid region of northeast
China (e.g., grasses), the $LUE$ remains approximately constant even for high values of $PAR(D)$ generated by aerosols and clouds (Jing et al., 2010). Overall, however, the $LUE$ is low for many vegetation types, typically between 1-3%.

### 3.5 The net absorption of CO$_2$ due to aerosols from fires

Figure 8 shows the relative changes in the $NEE$ during all months of the year, discounting confounding factors due to the seasonality, i.e., monthly changes of variables that strongly affect photosynthetic rates. Three essential reasons reinforce the
440 use of the whole year in these analyses: (1) wet season contains about 15-20% of the wildfires number detected during the dry season (Tab. S4). We observed numerous hotspots of fires around the area of the study, i.e., BBOA sources emitted locally and transported regionally (Fig. S5); (2) the relative contribution of BBOA during the wet season is relatively small but contributes to improving the sample space, considered a critical aspect to the study; (3) removing or maintaining transition and rainy periods in the analyses does not change the scientific direction of the results initially found in Figure 8b (see Figs. S11-12).

**3.5.1 Seasonality of biophysical factors on NEE**

To reduce the effect due to the seasonality of biophysical factors strongly driven by the change in weather conditions during the year (e.g., water stress, deciduousness, ecosystem respiration), we normalized Equation 11 by the clear-sky $NEE$ adjusted to each month of the year (Fig. S8) shows average monthly changes found in the period 2005-2008. These adjustments better support our assumptions regarding the derived quantities described in the methods. In fact, we observed a relative average

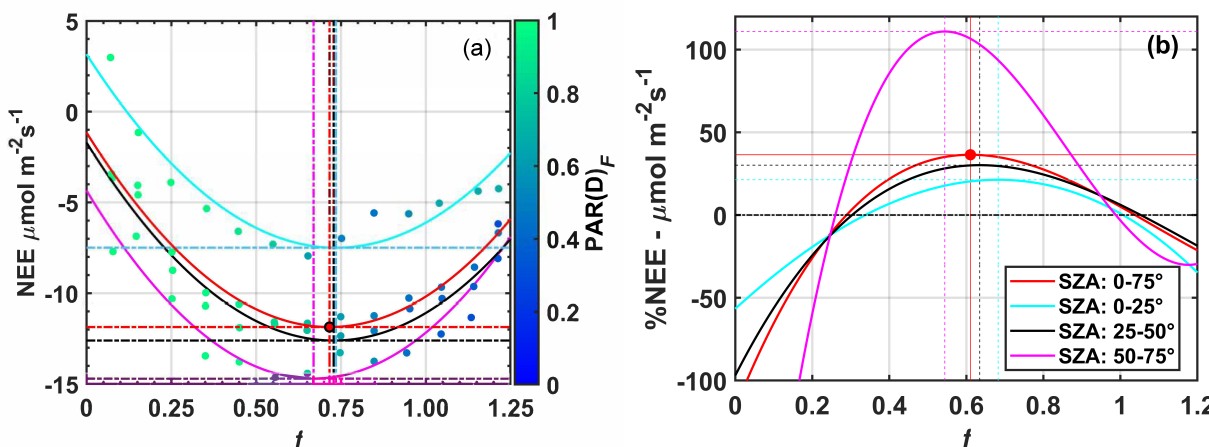

**Figure 8.** Variability of $NEE$ with $f$ for various $SZA$ ranges in (a). The $\%NEE$ as a function of the irradiance $f$ for the same $SZA$ intervals is shown in (b). The $\%NEE$ is calculated from Equation 11, corrected with the $NEE_0(sza)$ computed from the fit curves presented in Table S2. These graphs include the effects of aerosols in the experimental area of Cláudia-MT, between 2005-2008

increase of 30% on the $\%NEE$ to $SZA$ ranging from 0-75º after applying these corrections, compared with a single curve for all years. To SZA ranging 0-20º um enhancement up to 70% on the $\%NEE$ was observed. Since many studies do not take these corrections into account, these results suggest that the impacts of BBOA on the $NEE$ can be even more significant than previously known, especially in the Central (Manaus, K34) and Western Amazon (Ji Paraná, RBJ) (Oliveira et al., 2007; Cirino et al., 2014; Rap, 2015; Moreira et al., 2017).

The Equation 11 and Equation 4 allowed us to evaluate the behavior of the ratio between the $\%NEE$ and the irradiance $f$ for intervals $SZA$ from 0-75°. This procedure was adopted to minimize the effects of solar elevation and air temperature on the $NEE$ flux throughout the day (Gu et al., 1999; Cirino et al., 2014). The intervals every 25° ensured the smallest possible $SZA$ variations and the largest possible number of points within the sample space necessary for statistical analyses. For each $SZA$ interval analyzed, the average $\%NEE$ was evaluated in *bins* of $f$ equal to 0.1, calculated separately (Fig. 8). The critical points

and the coefficients of curves for all data (between 0-75° $SZA$) are shown in the supplementary material (Fig. S11, Table S3). On average, an average (absolute) increase of approximately 7.0 $\mu$mol m$^{-2}$s$^{-1}$ in carbon uptake was observed relative to clear sky conditions when $f$ varied from 1.1-1.0 to 0.66, results for the $SZA$ range between 0-75° (Fig. 8 (a). The 7.0 $\mu$mol m$^{-2}$s$^{-1}$ increase represents a 20-70% increase in $NEE$ flux. This increase, strongly linked to the increase in aerosol concentration by fires, is mainly explained by the 50% increase in $PAR(D)_F$, ($\approx$ 450 $\mu$mol m$^{-2}$s$^{-1}$ in the stream $PAR(D)$) and 35-40%

reduction in the irradiance $f$ when the AOD$_a$ varies from 0.10 to 5.0, As it was shown in the Figure 5b).

Oliveira et al. (2007) and Cirino et al. (2014) showed a relative increase of about 30% for $f$ values ranging from 1.1 to 0.80. These studies showed that the increase in carbon uptake in the presence of aerosols and clouds becomes smaller and similar in both locations for $SZA$ bands < 20. Solar radiation suffers less scattering near the zenith ($SZA \sim 10°$) due to particles suspended in the atmosphere and the narrowing of the optical path, reducing the effects of diffuse radiation on the

photosynthetic process. These results, in particular, are repeated for the studied semideciduous forest of Mato Grosso, but a strong increase of 70% in $\%NEE$ is observed for lower $SZA$ ranges (between 50-75%), in the early hours of the day, between 8-10h (LT), while in the Jaru Biological Reserve (JBR) the biggest increases are concentrated in the $SZA$ ranges between 10-35°, close to midday, or in the morning-afternoon (Oliveira et al., 2007). At K34, in Manaus, the maximum absorptions and the maximum $\%NEE$ that occur do not exceed 20% and the effects of aerosols and clouds operate together. The individual radiative influences of clouds and aerosols are difficult to quantify because satellite AOD observations have a low temporal resolution. Similar results were observed by Doughty et al. (2010) in FLONA-Tapajós, central Amazon. In general, higher standard deviations are found in regions most heavily impacted by aerosols (Oliveira et al., 2007; Cirino et al., 2014; Rap, 2015), such as Ji-Paraná (RO) and Alta Floresta (MT). Because aerosol concentrations are relatively lower in FLONA-Tapajós (PA) and Manaus (AM), the standard deviations are lower (Oliveira et al., 2007; Doughty et al., 2010; Rap, 2015). Theses deviations can be found in previous studies published by Oliveira et al. (2007) in FLONA-Tapajós (PA), Cirino et al. (2014) in Manaus (AM), and Ji-Paraná (RO).

Table 5 lists the coefficients of the adjustments found for $NEE$ and $\%NEE$ as a function of $f$ for each of the $SZA$ ranges considered. We identified the optimal and critical radiation conditions for carbon uptake between 07-17h (LT) and listed them below. As mentioned before, tipping points ($C_p$) represent the called physiological optimums. Our results show a substantial decrease (increase), significant statistically, of $NEE$ ($\%NEE$) as a function of $f$ from $-7.5$ $\mu$mol m$^{-2}$s$^{-1}$ to $-14.7$ $\mu$mol m$^{-2}$s$^{-1}$ and 1-0.63 ($\approx$40%) when SZA ranges from 0-25° to 50-75°, respectively ($R^2 \geq 0.85$).

**Table 5.** Polynomial adjustments (Fig. 8), coefficients, and statistics for the periods between 07-17h (LT) in the micrometeorological tower 50 km from Sinop-MT, in the municipality of Cláudia, between 2005-2008.

| Settings | Angles | Coefficients | | | | Statistic | |
|---|---|---|---|---|---|---|---|
| **Poly fit 2nd** | **SZA** | *a* | *b* | *c* | *d* | $R^2$ | $C_p\ (x_v, y_v)$ |
| | 0-25° | $+23$ | $-31$ | $-4.3$ | | 0.88 | (0.74, -07.50) |
| $NEE$ | 25-50° | $+21$ | $-30$ | $-1.7$ | | 0.95 | (0.73, -12.61) |
| | 50-75° | $+20$ | $-29$ | $+3.1$ | | 0.88 | (0.67, -14.71) |
| | 0-75° | $+21$ | $-30$ | $-1.1$ | | 0.97 | (0.72, -11.90) |
| **Poly fit 3rd** | **SZA** | *a* | *b* | *c* | *d* | $R^2$ | $C_p\ (x_v, y_v)$ |
| | 0-25° | $-3.5 \times 10^1$ | $-1.2 \times 10^2$ | $+2.1 \times 10^2$ | $-5.7 \times 10^1$ | 0.89 | (0.68, 21.31) |
| $\%NEE$ | 25-50° | $+1.4 \times 10^2$ | $-4.9 \times 10^2$ | $+4.6 \times 10^2$ | $-9.7 \times 10^1$ | 0.97 | (0.63, 30.13) |
| | 50-75° | $+1.1 \times 10^3$ | $-2.9 \times 10^3$ | $+2.1 \times 10^3$ | $-3.8 \times 10^2$ | 0.96 | (0.54, 110.9) |
| | 0-75° | $+2.0 \times 10^2$ | $-6.6 \times 10^2$ | $+5.8 \times 10^2$ | $-1.2 \times 10^2$ | 0.98 | (0.61, 36.40) |

These results correspond, respectively, to the relative increase ($\%NEE$) of about 25-110% in the first and last hours of the day, when the lower solar angles (greater physiological optimum), corroborated to results presented in Figure 5, Figure 6 and Table S4. For lower than 0.63, we also observed a strong decadent of photosynthetic rates until $f \sim 0.25$ when the photosynthesis process breaks altogether. Thus, $f \sim 0.66$ (SZA 0-75°) can be interpreted as a threshold for which photosynthetic rates ($NEEf^{-1}$) indicate a strong reduction in the carbon uptake capacity of forest in response to overload BBOA. As for enhancements in the NEE, attributed to light-shading, it must be sighted as a narrow resilience of forest in response to a polluted atmosphere. These results offer more direct insight into alterations in solar radiation caused by BBOA and its impacts on carbon uptake during the day, although there are uncertainties not measured associated with clouds.

## 3.6 Insights into ecosystem-respiration uncertainties

Since no direct local measurements of Ecosystemic Respiration ($R_{eco}$) exist, estimates are necessary. However, typically, the models available in the literature grossly overestimate or underestimate the local $R_{eco}$, especially when in-situ data are unavailable to fit them (e.g., autotrophic and heterotrophic respiration; litter, soil, trunks, branches, leaves, and roots) (Malhi, 2012). It is important to highlight that the indirect effects of BBOA on $R_{eco}$ were not exploited yet in the "Cerrado-Amazon Forest" ecotone. Little is known about how aerosols modify the $R_{eco}$ in the region. We highlighted that previous studies by (Vourlitis et al., 2002, 2011) made daily estimates for the $R_{eco}$ without isolating the radiation-attenuating effects due to aerosols. These conditions are pretty different for the current study. Once photosynthetic rates are also modulated by solar radiation (attenuated by BBOA), changes in the VPD may also be impacting ecosystem respiration from no-linear interactions, influencing the opening and closing of stomata, canopy temperature, humidity, and soil temperature. All these factors influence the microbiological dynamics of the soil and litter, with implications still unknown for $R_{eco}$ (in-situ). Thus, we assume that the uncertainties underlying the calculation of the $R_{eco}$ (for the reasons mentioned) could affect our results equally significantly by disregarding it. Therefore, we assumed that the temporal variability of GPP is similar to the temporal variability of NEE.

To better support our assumption ($GPP \approx NEE$), we have done a similar test (Figs. 8b and S11) using only daytime data during the dry season (Figs. S11 and S12). We found that the % change (on average) is similar, around 15% for $SZA$ (0-75°) and around 35% for $SZA$ (50-75°). Both differences may be attributed to uncertainties due to the daytime $R_{eco}$ (Fig. 9) and factors that need to be better explored in future work. We hypothesized some mechanisms that could lead to an increase in $NEE$ throughout the dry and smoky seasons (strongly supported by previous studies in the Amazon and world), as follows below. (1). During the dry season (Jul-Sep), photosynthetic deficit due to deciduality is partially compensated by positive feedback of extrinsic factors: BBOA concentration, $PAR(D)$, cooling of the air and leaf canopy, and VPD reduction (Vourlitis et al., 2001; Gu et al., 2003; Rap et al., 2018; Corwin et al., 2022). (2). During the flush new leaves season (Oct-May), photosynthetic enhancement is primarily explained by positive feedback both extrinsic (mentioned) and intrinsic factors (Leaf Area Index and $LUE$): variation in the characteristics of the forest canopy due to the newly sprouted leaves, i.e., higher photosynthetic capacity of canopy that compensates the unfavorable stomatal response due to precedent drought (Wu et al., 2016; Green et al., 2020); (3). The magnitude of the effects observed in assumptions (1) and (2) will strongly depend on the intensity and frequency of occurrence of meteorological phenomena (e.g., planetary limited layer dynamics) (Fuentes et al., 2016; Gao et al., 2021).

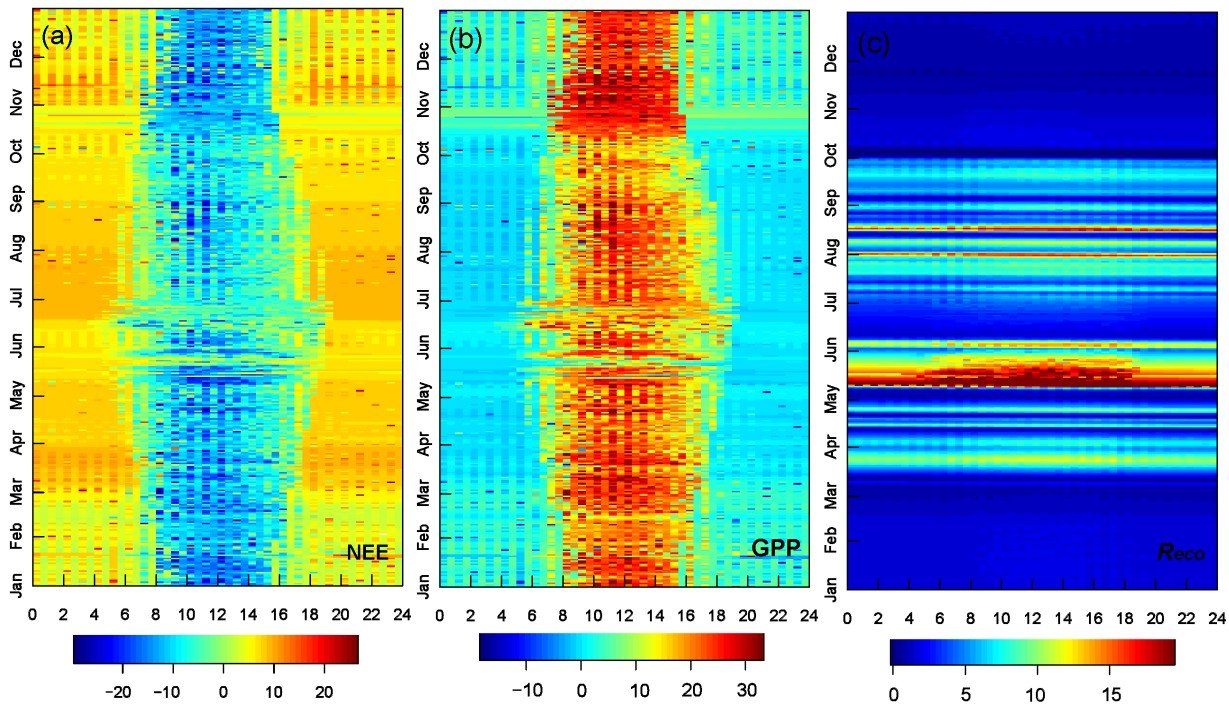

**Figure 9.** Seasonal changes on the fingerprints calculated by the REddyProc system in $\mu$mol m$^2$s$^{-1}$ during the years 2005-2008: (a) $NEE$; (b) $GPP$; and (c) $R_{eco}$ Daytime. The axis-x shows time (24h), UTC -4.

### 3.7 BBOA effects on the biophysical variables and NEE

These results are important as a large part of the Amazon area is frequently impacted by the presence of aerosols in small amounts (low AOD), similar to those observed in the north of the Amazon basin, in Manaus-AM. In regions with high rates of deforestation and biomass burning, however, increases in $CO_2$ absorption are significant and can have major impacts on the carbon budget of the Amazon forest. Over dense forest ecosystems of central Amazon, $CO_2$ absorption peaks are often observed at higher and narrower intervals of f (1.1 to 0.80), especially for dense forest ecosystems (Gu et al., 1999; Yamasoe et al., 2006; Oliveira et al., 2007; Doughty et al., 2010). This is different from grasslands and temperate forests, where the maximum net $CO_2$ uptake is generally found in the range $f$ between 1.0-0.5 (Gu et al., 1999; Niyogi et al., 2004; Jing et al., 2010; Zhang et al., 2010).

The mechanisms to explain the variation in $\%NEE$ with the irradiance $f$ are complex and influenced by the dynamics of the Planetary Boundary Layer (PBL) throughout the day, including transport of regionally transported and locally emitted burning emissions. For the semideciduous forests studied here, an accumulation of aerosols from fires during the night hours (19h to 06h, LT) maybe associated with greater stability in the PBL during the fire season (lower values in wind speed, reduction in convection, and boundary layer narrowing). These factors can increase the concentration of aerosols (AOD$_a$) during the night,

with important effects on the $CO_2$ absorption capacity ($\%NEE$) observed in the early daytime hours ($SZA$ values between 50-75°.

     Future studies may elucidate the dynamic effects of PBL on the photosynthetic capacity of forests in the Amazon Basin, like studies carried out in other forests around the world, e.g., in Utah, USA, Helliker and Ehleringer (2000), UK, Yakir (2003), and Beijing, China, Wang et al. (2021, 2022). Field experiments focused on the vertical distribution of $PAR(D)_F$ throughout

the canopy will improve the current understanding of the individual effects of aerosols and clouds on the forest microclimate ($LC_T$ and $VPD$) on $\%NEE$.

     Figure 10 shows significant interference of aerosols on environmental variables that consequently affect the photosynthetic dynamics of plants. The attenuating effect of incident solar irradiance due to the presence of aerosols triggers statistically significant reductions in $LC_T$, $T_{air}$ and $VPD$ near the forest canopy (Fig. 10). However, we noticed that the $T_{air}$ variability is

wider/broader than the $LC_T$ variability (Figs. 10b and 10a), which suggests that the $LC_T$ fails to capture the realistic variability. To verify and evaluate the consistency of the $LC_T$ model (Equation 9), calibrated to the Central Amazon conditions (mentioned in the subsubsection 2.3.7), a second method ($LC_{Ts}$), based on the Stefan-Boltzmann equation (Doughty et al., 2010; Cirino et al., 2014) was tested (see Equation S1, Figs. S9 and S10, supplementary material). We observed that $T_{air}$ is systematically smaller throughout the day (Figs. S9b and S9a), results obtained from the same data points shown in Figures 10a and 10b. On

average, the amplitude between $LC_T$ and $T_{air}$ is equal to 2.2 ($\pm$ 2.1) °C (Fig. S9c), and the leaf canopy is warmer than the air between 7-17h (LT), as expected. The leaf canopy was warmer than $T_{air}$ during the day. However, the standard deviation (std) of $LC_{Ts}$ is significantly higher. The amplitude between $LC_{Ts}$ and $T_{air}$ is about 1.7 $\pm$ 11.1 °C (Figs. S9a and S9b). $LC_{Ts}$ appear to capture average hourly behavior but exhibit much greater hourly variability compared to $T_{air}$ throughout the year. The $LC_T$ results reveal an acceptable average hourly pattern for leaf canopy temperature, although the pattern is unrealistic

compared to $T_{air}$. Due to the limitations of $LC_T$, it is worth mentioning that we are likely underestimating the uncertainties of aerosol effects on canopy temperature. However, the impact of aerosols in $T_{air}$ also indicates an important cooling at the surface ($\sim$3-4 °C), with relevant effects on the canopy and functioning of the studied ecosystem.

     Several mechanisms have been used to explain the increase in photosynthetic capacity by the canopy due to changes in the biophysical properties of the forest, among them, the general trend of decreasing temperatures (Koren et al., 2014; Bai

et al., 2012) and $VPD$ (Min, 2005; Yuan et al., 2019) under cloudy or smoky skies. The effect of this cooling, especially on the leaf canopy, can also exert considerable influence on the photosynthesis of the forest (Doughty et al., 2010; Vourlitis et al., 2011). Herein, the impact of aerosols produced, respectively, a cooling of 3 °C and 2.5 °C in the $LC_T$ and $T_{air}$ when $f$ declined from 1.10 to 0.66 (Fig. 10a and Fig. 10b), ranges for the which NEE increase about of 3-7 $\mu$mol m$^{-2}$s$^{-1}$ as discussed section before. These results are similar to the results found by Davidi et al. (2009) and Doughty et al. (2010) in the FLONA-

Tapajos (Santarem-PA). However, the individual impacts of these effects depend on long-term and simultaneous measures of extrinsic factors (water stress, nutrient availability, solar radiation, aerosols, and cloud cover) and intrinsic aspects of the plant (forest type, leaf canopy structure, stomatal and roots structure), unavailable at the site and period studied. Moreover, the non-linear relationship between these factors makes it challenging to determine the physiological optimums for given biophysical variables, such as temperature and VPD (Figure 10c).

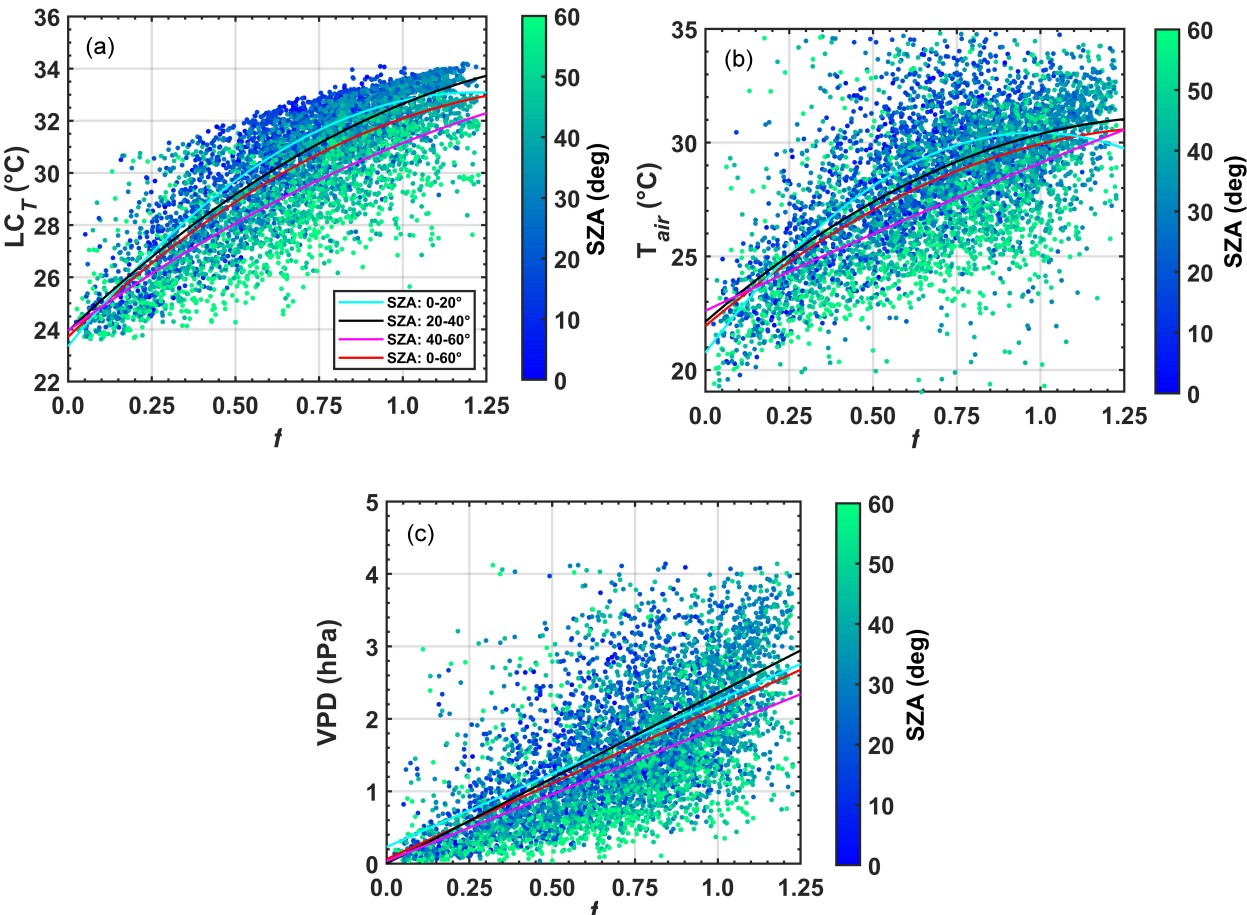

**Figure 10.** Correlation between the relative irradiance $f$ versus $LC_T$ (a), $T_{air}$ (b) and $VPD$ (c), values calculated for $SZA$ between 0 and 60. The air temperature was measured at 42 m above the ground in the micrometeorological tower located in the municipality of Cláudia, 50 km from Sinop-MT, using the parameterization given in Tribuzy (2005), between 2003-2004.

The increase in relative humidity due to air cooling induced by clouds or aerosols can also influence photosynthesis (Freedman et al., 1998; Altaratz et al., 2008; Jing et al., 2010). In many forest locations, the reduction in $f$ decreases $VPD$ during the dry season. These reductions, strongly influenced by the cooling of the air, are also closely linked with the cooling of the forest canopy and the increase in the absorption capacity of $CO_2$ ($\%NEE$) (Doughty et al., 2010), considering its physiological optimums (tipping points). For cloudy or polluted sky conditions, generally decreasing $VPD$ behavior can influence stomata
opening and intensify photosynthesis (Jing et al., 2010). Here, we observed a reduction of 2-3 hPa attributed to the decrease of around 3-4 (°C) in the air temperature, which agrees up to ∼40% reduction in $f$ from which NEE is critically reduced. Furthermore, It is still possible that an enhancement in the NEE is related to an increase in transpiration rates, providing cooling in the air, i.e., positive feedback between AOD, NEE, and $RU_{air}$ (Caioni et al., 2020).

Unlike what was found here, the forests of central Amazonia, in Manaus-AM (K34), FLONA-Tapajos (K83), Santarem-PA, and Ji-Parana (RO) seem to be less tolerant to the attenuation of sunlight-induced by clouds and aerosols. In our forest, the distribution of $f$ is close to 0.66 for $AOD_a \gg 0.10$ Table 5. This value is 15-20% lower than values found in central Amazonia when the $NEE$ reaches maximum negative values during the burning season ($f \sim 0.80$). This is the threshold value at which maximum carbon absorption is observed due to aerosol load in the JBR in the Ji-Parana JBR (south of the Amazon basin) as well as in the Cuieiras reserve at K34, in Manaus-AM. These comparisons are relevant because higher (lower) amounts of aerosols and clouds in the Amazon region can cause certain types of forests to absorb even higher (lower) amounts of carbon throughout the day (Gu et al., 1999; Cirino et al., 2014). The %$NEE$ frequency distribution patterns and their impacts on photosynthesis remain unknown for many other forest types in the Amazon and around the world. The results reported here are also consistent with calculations by Gu et al. (1999) for temperate forests in Canada, where negative maximums in $NEE$ flux occur for ranges $f$ between 0.55– 0.60.

The interannual variability of the relationship between the observed $AOD_a$, fire counts, and $NEE$ could not be analyzed, mainly due to the lack of a long-time series of $NEE$ flux data in the region. In the central Amazon, significant variability was observed from year to year. Higher %$NEE$ were often found on days with high fire counts. However, water stress and nutrient availability also play an essential role in the carbon uptake capacity (Gatti et al., 2014; Hofhansl et al., 2016; Gatti et al., 2021; Malhi et al., 2021). Joint modifications in these variables make it extremely difficult to quantify the individual effects of aerosols and clouds on the $NEE$. Field experiments taking measurements of all these aspects will yield studies with more robust and comprehensive conclusions on the ecosystem responses of Amazonian forests to external environmental disturbances such as fires.

## 4   Conclusions

The aerosol optical depth derived from the AERONET system proved to be a key variable in the elaboration of the clear sky solar irradiance model used to determine the relative irradiance $f$. The conceived model can be directed to other regions of the Amazon as long as they are within the same latitude range, where there are no $SW_i$ measurements. The parameter $f$, allowed us to satisfactorily evaluate the radiative effects of aerosols from fires on the net absorption of carbon ($NEE$) by the studied semideciduous forest ecosystem. The radiative impacts on $PAR_i$ and $PAR(D)$ allowed us to evaluate the impacts on the canopy light use efficiency ($LUE$), which increased by $\sim$ 1-3% under polluted conditions ($AOD_a$). The changes in incident solar radiation and $CO_2$ flux ($NEE$) could be attributed to the combined effects of aerosols emitted locally, regionally, or transported from more distant regions, considering the applied methods.

In the studied semideciduous forest ecosystem, the ($NEE$) increased from 20-70% when the optical depth varied from 0.1 to 5.0 (on average). This effect was attributed to an average reduction of up to 40% in the amount of total PAR radiation, and also to an increase of up to 50% in the diffuse fraction of radiation ($PAR(D)_F$). This increase in $CO_2$ absorption capacity by the ecosystem is closely linked to the floristic composition of the understory and certain types of forest species adapted to low light conditions, which consists of more efficient vegetation in capturing diffused light during the photosynthesis process. The results

show higher photosynthetic efficiency under smoky sky conditions; loaded with particles scattering solar radiation due to fires, but also reveal the maximum limit in the PAR radiation cuts required for the photosynthesis process. Relative irradiances $f$ less than 0.66, on average, indicate the critical point at which forest photosynthetic rates undergo drastic reductions. Relative irradiance values $f \sim$ of 0.22 indicate 100% interruption in the photosynthetic process.

Due to the increase in the concentration of aerosol particles from fires in the region, statistically, significant changes were also observed in meteorological (biophysical) variables such as leaf canopy temperature and $VPD$. Scientific findings reveal a strong influence of fire aerosols on these variables, with potentially important effects on photosynthesis and carbon absorption. The 3 and 5 °C reductions in leaf canopy and air temperature are strongly associated with a 40% reduction in $f$ and a $\sim$ 2.0 hPa reduction in $VPD$ values which induce opening stomata and contribute to the observed increase of 20-70% in the $CO_2$ absorption capacity of the forest (%$NEE$). The individual influences or contributions of the $VPD$, $T_{air}$ and $LC_T$ to the ecosystem's net balance of $CO_2$, however, could not be directly quantified in this research. Indirect correlations, however, reveal statistically significant effects between the mentioned biophysical variables and the observed changes in the $NEE$ flux during the exposure of forests to fire and high values of $AOD_a$ (greater than 1.25, on average). Studies focused on the impacts of fires on the flux of water to the atmosphere deserve attention and can help to understand the role of forests in maintaining rainfall and its effects on the hydrological cycle (studies not yet carried out for most biomes in the Amazon).

## 4.1 Suggestions for future work

A more comprehensive regional study of the effects mentioned here, based on other vegetation types and biomes, using vegetation maps, remote sensing estimates, meteorological data, and numerical modeling, will help to better understand how the climate and ecosystem function in the Amazon are affected by natural and anthropic environmental. The reductions in the $NEE$ flux and, therefore, the reduction of the photosynthetic capacity of the plants due to the excessive increase in the concentration of BBOA aerosols and drastic reductions in the fluxes of solar radiation ($f \leq 0.22$) due to the fires in the region, constitutes an effect of notable relevance for carbon cycling in semideciduous forest environments in the Amazon and, therefore, an important contribution to a better understanding of this cycle in the region. Future work would also involve estimating the global-scale significance of aerosol load on Photosynthetically Active Radiation (PAR) and, consequently, on Net Ecosystem Exchange (NEE) fluxes. In addition to the potential use of numerical modeling, promising approaches in this direction include research on remotely sensed solar-induced chlorophyll fluorescence (SIF) (Meroni et al. 2009). SIF has been increasingly utilized as a novel proxy for vegetation productivity. Comparing SIF with remotely sensed $PAR_i$ and $PAR(D)$ (e.g., Rui et al. (2022); Zhang et al. (2023)) enables an observational-based global-scale comparison of the effects observed in this study and, potentially, the estimation of the impact of aerosol load in general, and biomass burning in particular, on the photosynthetic capacity of plants.

*Data availability.* This section provides free access to data repositories that support the conclusions. We provide the data from this survey available through the Mendeley Data platform (https://data.mendeley.com), where we will make upgrades and possible corrections. Citation: Cirino, Glauber; Vourlitis, George; Silva, Simone; Palácios, Rafael (2022), "Brazil-FluxMet-Stf", Mendeley Data, v2 DOI: 10.17632/m5h5fw872g.2. Secondary data is already in the public domain. We have listed the links to these data in the Supporting Information (Table S5)

*Author contributions.* Conceptualization and Methodology, S.R., G.C. and G.V.; Software, S.R., G.C., G.V. and R.P; Validation, G.V., G.C., R.P. and S.R.; Formal Analysis, The authors contributed equally to this work; Investigation, S.R., G.C., D.M., A.P., SC.L,B.I. and G.V.; Resources, G.V. and G.C.; Data Curation, G.V., J.N., G.C., R.P., and S.R.; Writing-original Draft Preparation, S.R. and G.C.; Writing-review and Editing, G.C., S.R., G.V., D.M., A.P., MI.V. and SC.L.; All authors have read and agreed to the published version of the manuscript.

*Competing interests.* No potential conflict of interest was reported by the authors.

*Acknowledgements.* We want to thank the Coordination for the Improvement of Higher Education Personnel (CAPES), the National Science Foundation, the National Council for Scientific and Technological Development (CNPq), the Mato Grosso State Research Support Foundation (FAPEMAT), the State University of California, San Marcos (CSUSM), to the Federal University of Mato Grosso (UFMT). Additional funding was provided by CNPq Universal, project 422894/2021-4, and by the Fundação de Amparo à Pesquisa do Estado do Pará (FAPESPA), grant 2022/45107. To PROPESP/UFPA, for the financial support to students. Special thanks to Dimensions Sciences Bridging Gaps with Scholarships (DS BRIDGES) - Amazon Task Force for providing social and financial support for the research of countless students in vulnerable situations. Our special thanks to Professor Dr. José de Souza Nogueira *(in memoriam)* who worked with other collaborators to generate and obtain micrometeorological data from the measuring tower used in this research.

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
