# Peer review of "Enhanced net CO2 exchange of a semideciduous forest in the southern Amazon due to diffuse radiation from biomass burning"

_EGUsphere, 2023_

## Author Comment (AC1)

https://doi.org/10.5194/egusphere-
2023-684 Preprint. Discussion started: 27
April 2023 Author(s) 2023. CC BY 4.0

[Figure]

**Final Response**
**egusphere-2023-684**
**Rodrigues et. al (2023)**
**simone.silva@ig.ufpa.br**

[Figure]

**Responses and actions on referee comments on the manuscript egusphere-2023-684, Enhanced net CO$_2$ exchange of a semi-deciduous forest in the southern Amazon due to diffuse radiation from biomass burning, by Simone Rodrigues et al., 2023.**

Date: July 15, 2023

**Authors' General Comments:**

We are deeply grateful to both reviewers for all their comments on our manuscript. We accepted all suggestions for corrections and changes that helped to make the final version of the manuscript clearer and better. Not only that, but we've done a careful review and removed all inconsistencies pointed out by reviewers. We are finalizing and adding some figures as suggested. Likewise, We will address all individual comments from the two reviewers and clarify our actions on each critique and suggestion. We'll start with the responses from Referee #1, followed by Referee #2. Our answers are just below each comment (in blue color). Some graphical analyzes have already been done and are shown below, others, such as GPP vs. $f$ and %GPP, are being processed, as they depend on acquiring new data. We hope to get the manuscript completely revised in a few days (our final version to referees #1 and #2), but we are also willing to insert new analyzes if necessary to improve the quality of the article further.

We also want to highlight two relevant points of this research (novelty and benefits): — (1) our most important scientific issue refers to the responses of transition forests to the aerosols and their impacts on the carbon uptake capacity at the ecotone Cerrado-Amazon Forest. As far as we know, this is the first study to report these effects in the transition between the Amazon and Cerrado biomes, which are considered critical in central Brazil. The novelty was analyzing the impact of (BBOA) emissions on the NEE in a semideciduous forest ecosystem in the southern Amazon basin from the local measurements (in-situ). The manuscript highlights the inflection points (critical points) from which the photosynthetic process drastically decays in the studied ecotone. We also developed a clear-sky solar irradiance algorithm based on the long-term measurements from the AERONET system, adjusted to the environmental conditions in the region concerning the vertical profile of trace gases (attenuators). However, while it is a useful by-product, it was not our focus. — (2) our results can improve the Dynamic Vegetation Modelling (DVMs) in these ecotones. In addition, they provide useful information for global public policies for preserving pan-tropical ecosystems in the face of the impacts caused by climate change, intensified by no-climate factors, such as anthropogenic actions.

**Referee Comments #1**

**General comments:**

The carbon cycle in the Amazon has been directly and indirectly impacted by both climate change and land use/land cover change, but the effect of each driver on net ecosystem exchange (NEE) remains uncertain. In this study, Rodrigues, Cirino and colleagues seek to use a combination of observations from an eddy covariance tower, and aerosol optical depth estimates from in situ and satellites to assess the impact of biomass burning aerosols on the radiation partition (direct/diffuse) and how it cascades into the carbon fluxes in a semi-deciduous forest site in the southern part of Amazon.

The topic is well suited and relevant for Biogeosciences, and the authors have data and analytical tools to provide an important contribution. However, the current analyses have important assumptions that are not clearly evaluated, and potential confounding effects are not addressed or discussed. In addition, at many places in the Results and Discussion section, the authors make statements that are not clearly supported by the results. I list some of these points below. Most of these concerns are fixable, but it will likely require substantial revision of the methods, analyses and discussion.

Most of the analyses presented here assumes that the variability of NEE is driven by gross primary productivity (GPP), and most of the variability in light quality is driven by aerosols. Ecosystem respiration, which had been pointed out previously as the main driver of seasonal variation of NEE in the same area (Vourlitis et al. 2011) is not mentioned as a potential confounding effect. More generally, if the goal of the research is to assess the effect of smoke on diffuse light and NEE, I wonder why the authors did not restrict the analysis to the dry seasons of the study period. That would likely reduce confounding effects due to seasonality (e.g., water stress, deciduousness, ecosystem respiration) and potentially provide better support for most of the assumptions in the derived quantities described in the methods.

**AC:** We insert some clarifications below regarding assumptions not discussed or (clearly evaluated). As for the unsupported claims, we are working to generate additional results that might support them and will insert some discussion sentences about them (in the revised text). E.g., We've worked on the $GPP$ preliminary tests and observed consistent and similar results compared to Figure 8b (preprint version). The procedures used in calculating the $GPP$, we'll put them in the supporting material or inserted in the main body of the manuscript. Regarding potential confounding effects, additional results/graphical have been added (corrected results below). We've calculated fit curves – $NEE_0(asz)$ – for each month of the year along the dataset and removed/minimized the effects due to seasonality, as suggested/appointed. From these corrections, it observed an increase of about 30% in the %$NEE$ (Figure 6). The corrections indicated that the $NEE_0(asz)$ must be adjusted as a function of seasonally to avoid sub-estimates in the %$NEE$. More details about the issues raised are in the sections below. We also want to highlight that the results reached from these suggestions enriched the quality of the manuscript.

**Specific points - RC#1**

**1**. Leaf canopy temperature (Section 2.3.7). The model used to estimate this quantity seems to come from Tribuzy et al. (2005), and applied to the focus site in Mato Grosso. Interestingly, Equation 9 does not depend on air temperature. Perhaps this is less of an issue for Manaus, the equatorial site where this equation was originally developed. However, as the authors indicate, Mato Grosso does experience temperature variations from weather systems. Moreover, in Figure 9, the air temperature range is broader than canopy temperature, which suggests that the modelled canopy temperature fails to capture the actual variability. I understand the authors do not have any validation data, but they should consider this limitation when using the canopy temperature estimates".

**AC:** In the current version of the manuscript, we have enhanced the text with new sentences to clarify the limitations of the $LC_T$ model. We agree that $T_{air}$ is broader than $LC_T$. However, $T_{air}$ is systematically smaller throughout the day (Figure 1a and Figure 1b) below, results obtained from the same dataset as Figure 9. On average, the amplitude between $LC_T$ and $T_{air}$ is equal to 2.2 ($\pm$ 2.1) °C (Figure 1c), and the leaf canopy is warmer than the air between 7-17h (LT), as expected. However, to verify the issue appointed and evaluate the consistency of the $LC_T$ model (Tribuzy, 2005), a second method was tested

($LC_{Ts}$) based on the Stefan-Boltzmann equation (Doughty et al., 2010; Cirino et al., 2014). We observed similar behaviors compared to Figure 1, i.e., leaf canopy warmer than $T_{air}$ during the day. However, the $LC_{Ts}$ standard deviation (std) is significantly higher. On average, the amplitude between $LC_{Ts}$ and $T_{air}$ is about 1.7 ($\pm$ 11.1) °C Figure 2. $LC_{Ts}$ seems to capture the average hourly behavior but gives a grossly greater variability hourly compared to $T_{air}$ (7-17h, LT) during the year. Both Figure 1 and Figure 2 can be inserted in the supplementary material as Figure S7 and S8 (if necessary). From our perspective, the $LC_T$ results reveal an acceptable average hourly pattern for leaf canopy temperature, although the std is unrealistic when compared to $T_{air}$. Due to the $LC_T$ limitations, we are probably underestimating the uncertainties of aerosol effects on canopy temperature. However, the impact of aerosols on $Tair$ also indicates an important cooling at the surface ($\sim$ 3-4 °C), with relevant effects for the canopy and functioning of the ecosystem studied. Therefore, we've kept the original Figure 9 in the body of the manuscript and briefly discussed the limitation when using the estimates $LC_T$, as suggested. The $LC_{Ts}$ model below (Equation 1) can also be inserted in the support material, as Equation S1.

$$LC_{Ts} = \sqrt[4]{\left( \frac{SWi(1 - \alpha + LWa - Rn)}{\varepsilon\sigma} \right)} \tag{1}$$

Where: **Rn** is the net radiation measured in the experimental area (Wm$^{-2}$); $\alpha$ is the mean albedo of the leaf canopy during the dry and wet seasons, respectively equal to 0.079 and 0.126 (Marques et al., 2017); $LW_a$ is the atmospheric longwave radiation in Wm$^{-2}$ (Idson and Jackson, 1963), adjusted for the polluted atmospheric conditions of Mato Grosso during the dry season (von Randow et al., 2006); $\epsilon$ is the emissivity, assumed to be 0.98 (Monteith, 1990; Marques et al., 2017) and, $\sigma$, the Stefan-Boltzmann constant ($5.67 \times 10^{-8}$ Wm$^{-2}$ K$^{-4}$). A method similar to this has been widely used, lacking direct measurements of leaf temperature in the Amazon (Doughthy et al., 2010; Cirino et al., 2014; Aguiar et al., 2012; Andrade et al., 2021).

**2**. Clear-sky NEE (Section 2.3.8). I wonder about what this equation is actually capturing. The same SZA may mean different times of day (and temperature and VPD) at different times of the year, so water and heat stress and deciduousness may be adding time-dependent noise on the fitted model (Eq. 10). Perhaps the authors should include month as a factor in their models? Or eliminate seasonal confounding effects by focussing on the dry season only?

**AC:** An excellent point was raised! To remove confounding effects, we recalculate $NEE_0(asz)$ using a curve for each month of the year Figure 3. Curve coefficients are listed in Table 1. We estimated the mean hourly $NEE_0(asz)$ cycles from Table 1 to better illustrate the differences concerning the mean profile to the twelve months of the year originally used in the manuscript (Figure 3). Figure 4 shows relevant variations and suggests relevant corrections in the $\%NEE$ calculation (Figure 8b, manuscript). We observed an average increase of about 30% in $\%NEE$ after corrections were applied (see Figure 5 and Figure 6). Based on these results, we decided to remake Figure 3 and Figure 8b of the manuscript. We understand that these procedures corrected/minimized the identified confounding effects. Therefore, we decided to keep the rainy and transition months in our analyses between 2005-2008. Three additional essential reasons reinforce our decision: (1) during the wet season, there are isolated fires in the study region, i.e., BBOA sources locally emitted and regionally transported from elsewhere; (2) the relative contribution of fires during the wet season is small; about $\sim$10% of the sample space, but it still contributes to improving the statistics which is a critical aspect of the study; (3) removing or keeping transitional and rainy periods in the analyses, does not change the scientific direction of the results originally found in Figure 8b and the underlying conclusions (see Figure 5 and Figure 6).

**3**. Results and discussion section. I found this section difficult to follow, and often found myself unsure on whether the authors were describing their results or previous research. If the authors prefer to keep the results and discussion together (as opposed to separate sections), I suggest reorganising the paragraphs so the distinction between results from this research and the discussion within the broader literature is very clear. For example, in Section 3.1, it was difficult to separate which results were from this study and which results came from Vourlitis et al. (2011). Perhaps focus more on describing Figure 4 and only briefly compare/contrast with the previous results by Vourlitis et al. (2011).

**AC:** Okay, done. We have reorganized the paragraphs, giving greater attention to our results and briefly discussing the similarities in the wider literature. Likewise, we updated the paragraph of Figure 4 according to suggested. The updated discussion, it'll follow with the revised manuscript.

**4**. In addition, the authors used NEE and GPP interchangeably throughout most of the manuscript (including in the definition of light use efficiency). Within the same season and during daytime, I can see that this is less likely a problem, but unless I missed this, the authors used the time series across all seasons, which could confounding effects. I understand that GPP estimates from eddy covariance flux towers can be uncertain too, but I think the authors could explain why they opted for analysing NEE instead of GPP. Also, the authors highlight that the forest is semi-deciduous, which made me wonder about the mechanisms that could lead to an increase in NEE through increase in GPP during the dry and smoky season. The assumption of GPP-driven variability in NEE appears multiple times (e.g., L439-L457, L.494-502), so it seems central to the discussion, yet it is not fully supported by the presented data and analyses.

**AC:** We are working on that. Since no direct local measurements of Ecosystemic Respiration ($R_{eco}$) exist, estimates are necessary. However, generally, these equations greatly overestimate or underestimate the local $R_{eco}$, especially when in-situ data are unavailable, i.e., autotrophic and heterotrophic respiration (e.g., litter, soil, trunks, branches, leaves, and roots). As the $R_{eco}$ is an important factor throughout the year, we reinforce that we do not use the NEE as an approximation of the *GPP*. Although this does not seem evident throughout the article (we will highlight in the revised manuscript). Herein, we assume that the uncertainties underlying the calculation of the $R_{eco}$ (for the reasons already mentioned) could affect our results equally importantly when disregarding it. Thus, we neglected $R_{eco}$ to preserve our analyses from systematic errors, i.e., equally or even more significant uncertainties, by eliminating it from the equation $GPP = NEE - R_{eco}$. We also highlight that Vourlitis et al. (2011) made daily estimates for the $R_{eco}$ without isolating the effects of pollution from fires. These conditions are quite different for the current study. As this study is a pioneer in the "Cerrado-Amazon Forest" ecotone, we still do not know how aerosols modify the region's $R_{eco}$ (direct or indirect). Since photosynthetic rates are also modulated by solar radiation (attenuated by local aerosols), changes in the VPD may be impacting ecosystem respiration from no-linear interactions, influencing the opening and closing of stomata, variations in canopy temperature, humidity, and soil temperature. These factors strongly influence the microbiological dynamics of the soil and litter, with implications still unknown for $R_{eco}$ (in-situ). However, we are willing/curious to see how much our results change when considering it. To this end, we're testing the "REddyProc" model, available in R code by the team of Max Planck Institute for Biogeochemistry (BGC, Jena/Germany). We are still working on these analyses but are committed to finalizing them in a few days. We hope this is not an issue while reviewing the article.

**5.** Likewise, the authors discuss the effects of potential confounding effects on the observed relationship between diffuse light and NEE in Section 3.6 and in Figure 9 (e.g., temperature and vapour pressure deficit), but they do not account for these other important drivers in their analyses. They mention in line 508-509 that they could not quantify the effect of these variables in this study, but they do not explain why, and I do not see any reason for not exploring it with statistical models that account for these other variables (similar to the models that they implemented, but with additional predictors). The authors did attempt to mitigate these effects by exploring the response of NEE to diffuse radiation fraction by binning data by solar zenith angle (Section 3.5, Figure 8), but that may have caused the bins to have different times of day grouped together across seasons, so it is difficult to interpret the results.

**AC:** It is a good point raised. We have two paths to follow: (1) atmospheric and canopy radiative transfer models coupled (e.g., Dynamic Vegetation Models – DVMs), or (2) our fits to try to separate and estimate the suggested effects. From our standpoint, (1) it's a little outside the scope of this study, and (2) it is doable as long as 1–2 weeks is granted to us by the editorial. Anyway, we've improved our discussions as suggested. We added a brief paragraph to discuss aerosols' multiple effects on the biophysical variables that modulate photosynthesis rates.

**Minor points - RC#1**

I found the text to have several typographic mistakes and sentences that appear out of place. I am not listing every one, but I suggest the authors to thoroughly revise and streamline the revised text for clarity.

**AC:** Thank you for reviewing the article in detail. We've reviewed the notes highlighted below and corrected all points noted. In addition, we thoroughly revised the article for gross typographical errors and misplaced sentences, as noted.

**1.** L1. The opening sentence is a bit circular (atmospheric processes and climate are closely linked to carbon cycle as a

consequence of biosphere-atmosphere coupling).

**AC:** Ok, we have replaced the sentence with carbon cycling in the Amazon fundamentally depends on the functioning of ecosystems and atmospheric dynamics, which are highly intricate.

**2.** L2. The radiative effects of aerosols and clouds ON XYZ are still unknown…

**AC:** Ok, replaced for: The radiative effects of aerosols on the Amazon-Cerrado biomes still need to be discovered for a wide variety of vegetation, usually unconsidered since that was not inventoried in previous studies.

**3.** L5. Relative irradiance: briefly explain "relative" (i.e., relative to which conditions)

**AC:** We agreed and rewritten it as follows: Our results show a decrease in incident solar radiation of up to 40% in regard to smoky sky conditions (i.e., a decrease in relative irradiance $f$ from 1.10 to 0.67). Consequently, we observed an average increase in the carbon uptake ($\%NEE$) of 20%, 40%, and 110% for $SZA$ ranging between 0-25°, 25-50°, and 50-75° (deg), respectively.

**4.** L10. 10% increase or 10% decrease?

**AC:** We have revised and rewritten it: "… Important influences on VPD and air temperature and canopy induced by the interaction between solar radiation and high aerosol load in the observation area were also noticed. On average, an increase up to 2-3 hPa and a cooling of about 3-4 °C is observed, respectively…"

**5.** L17-20. The nature of the debate is unclear, consider briefly explaining it.

**AC:** Ok, we have added sentences to explain the nature of the debate, and we made it clearer as below: The role of Amazonian Forest ecosystems has been widely debated in the context of global climate change. Redistribution of biomes and plant species (Davison et al., 2021), loss of biodiversity (Brando et al., 2014; Saatchi et al., 2021), increase in fires (Brando et al., 2019; Alencar et al., 2022; Sullivan et al., 2020), outbreaks of pests and diseases (Anderegge et al., 2020) are examples of impacts, aggravated not only by climatic factors but also by anthropogenic ones (Ometto et al., 2022). These impacts have been threatening the largest pantropical $CO_2$ sinks since 1990'. Reductions from 1.26 PgC yr$^{-1}$ to 0.29 PgC yr$^{-1}$ are expected between 1990-2030, possibly reaching zero in the Amazon (Hubau et al., 2020).

**6.** L24-25. Mention other sources of CO2 too? Deforestation and degradation (including fires).

**AC:** Thank you for the suggestion. We have inserted other sources in the body text as follows: $CO_2$ absorption through photosynthesis increases the vegetation and soil C stocks, representing a C sink, while plants, animals, microbial respiration, decomposition of dead vegetal biomass, and wildfires release $CO_2$, representing a C source to the atmosphere (Artaxo et al., 2022; Venturini et al., 2022, Silva Junior et al., 2020)

**7.** L36-39. I found this discussion somewhat misleading due to the significant difference in scale across the studies (Gatti et al. is a regional study, whereas the other references are for specific sites).

**AC:** Ok, we rewrite the sentences as below: However, regional numeric modeling (Moreira et al., 2017) and in-situ studies indicate (Carswell et al., 2002; von Randow et al. (2004) that Amazonian forests can occasionally be net atmospheric $CO_2$ sinks; or approximately in equilibrium (Vourlitis et al., 2011).

**8.** L55. The sentence is vague. What are the current limitations are why do these limitations matter?

**AC:** Ok, we agreed. We reinforced the text with info about the current limitations and its matter, as follows: The models, however, need improvements in parameterizing the radiative effects of aerosols and clouds on the $NEE$, e.g., a more realistic representation of the canopy structure and processes leaf physiological and morphological (Duran et al., 2021). Improvements in the aerosol optical model, its properties, secondary formation, lifetime, evolution, and absorption of aerosols are also critical (Drugé et al., 2022), especially those related to shape, size, and chemical composition. These

improvements are fundamental for a more accurate and realistic spatial distribution of the atmospheric $CO_2$ absorption potential by Amazonian forests (Procópio et al., 2014; Moreira et al., 2017).

**9.** L57-59. Doesn't Rap et al. (2015), which the authors already cite, discuss the effects of aerosols on productivity across the Amazon (including Mato Grosso) using numerical modelling?

**AC:** Rap et al. (2015; 2018) have studied the effects of aerosols on productivity across the Amazon Basin, considering only numeric simulations (DVGMs). The specifics of semideciduous forests have been unconsidered in both studies.

**10.** L76. Remove "105"? It seems out of place.

**AC:** Ok, done.

**11.** L84–85. The areas presented in this sentence (49.95 km2 and 20.50km2) seem very small for Mato Grosso.

**AC:** We've corrected and updated these numbers and the citation as below: The areas of transitional forests (Amazon Forest-Cerrado) covered approximately 41% (362,538 km²) of the State of Mato Grosso. Due to the advance of the agricultural frontier, 21% of these areas suffered drastic reductions. Part of these forest areas are found in protected areas and territories of indigenous communities (approximately 17%). The deciduous and semi-deciduous forests of the Cerrado biome initially covered 49,951 km² in the State of Mato Grosso. Deforested areas represented about ≈ 41% of this total, with only 14% located in conservation units in Alencar, etc. (2004). The geographic positions of these forests are discontinuous due to climatic fluctuations that have occurred in the last 10,000 years Prado and Gibbs (1993).

**12.** L94. Which systems operate in northern Amazon?

**AC:** Ok, we have mentioned the weather systems in the area of study as follows: This area's 30-year average annual temperature is 24°C, with precipitation of approximately 2000 mm yr$^{-1}$ (Vourlitis et al. 2002). The Bolivian High (BH) and South Atlantic Convergence Zone (SACZ) are among the active atmospheric systems in the northern Mato Grosso region, while the southern is affected by extratropical systems, such as Frontal Systems (Reboita et al., 2012).

**13.** L97. Flush new leaves? "Recover" strikes me as the incorrect word, as deciduousness is an evolutionary adaptation to droughts.

**AC:** Ok, we have replaced the word below: The loss of leaves (deciduousness) during the dry season (July-September) is quite sensitive to water availability and temperatures (maximum and minimum) in the region. With the arrival of the wet season (November-May), flush new leaves occur with typical characteristics of tropical forests (Vourlitis et al. 2011).

**14.** Section 2.2.1. What is the time span of the AERONET data, 1993-2018 (L105) or 1993-2021 (L121).

**AC:** Ok, done. The data period has been corrected in the manuscript for 1999-2017.

**15.** L143. Drop "in Amazonia" as eddy covariance has been used globally.

**AC:** Ok, done. Replaced to: The eddy covariance system has been widely used to measure the net $CO_2$ flux by the ecosystem.

**16.** L148-149. Drop sentence? This does not seem to add much content.

**AC:** Done. That sentence was deleted.

**17.** L155-157. Sentence is confusing.

**AC:** Ok, we rewrite the sentences as below: The $NEE$ is obtained from the eddy-covariance system. The eddy system provides $CO_2$ flux measurements at 10 Hz from a sonic anemometer integrated. For $NEE$ calculation, the storage term

S[$CO_2$] is obtained according to Aubinet (2012) and Araújo et al. (2010). For S[$CO_2$] term calculation, we considered continuous measures of the $CO_2$ concentration vertically arranged between the ground and the top of the tower (Vourlitis et al., 2011). Under these conditions, the $NEE$ of $CO_2$ is approximated by Equation 1:

**18.** L170. Temperature should be in K, not °C, for equation 2.

**AC:** Ok, done. We have verified and replaced °C with K as noticed.

**19.** L189. What is Meteoexploration (SolarCalculator)? Provide reference/citation/context.

**AC:** Ok. We inserted three references and clarified the text. The updated sentences are as below: The Solar-Calculator is a free system used to compute the clear sky solar irradiance, managed by Meteo Exploration company. The solar irradiance is calculated according to Bird and Hulstrom (1981b), updated by Corripio (2003). The hyperlinked for Solar-Calculator is listed in Table S3.

**20.** Equation 4. The notation is somewhat confusing. Perhaps replace the numerator with SWia(t), so it is universal (as opposed to only when AODa > 0.10 and accounting for cloud cover)?

**AC:** Ok, we change the numerator of Equation 4 as suggested and insert the typographical value $SW_i(t)$ throughout the manuscript. Moreover, we revised all typographical values in the text body. Equation 4, updated, is below:

$$f = \frac{SW_i(t)}{S_0(t)\,\{AODa < 0.10, cloudless\}} \tag{2}$$

**21.** L244-245. The definition of LUE reads as a bit too circular to me.

**AC:** We have redrafted the wording of the ($LUE$) definition as follows: "… Another important parameter in this study is the light use efficiency ($LUE$). Herein, the ($LUE$) is an estimation of the efficiency with which plants convert solar radiation (PAR) into chemical energy through photosynthesis (Gonsamo, 2018) …" We also have recalculated the ($LUE$) based on $GPP$ (Equation 8), updated below:

$$LUE \cong \frac{GPP}{PAR_i} \tag{3}$$

**22.** L281. List the bins used?

**AC:** Ok, we indicate in the text the $SZA$ ranges. We used $SZA$ intervals every 25 (deg), considered statistically acceptable for the data set. We also performed analyses with smaller intervals (e.g., bins every 20 and 15 deg) but found few or no points in some bins. A similar procedure has been widely used (Gu et al., 1999, 2003; Oliveira et al., 2007; Cirino et al., 2014).

**23.** L290. Elaborate and briefly describe/provide examples of what were the acceptable levels?

**AC:** We have briefly elaborated and described a few sentences with acceptable levels examples. The sentences inserted are below: We exclude unexpected maximum and minimum values for the region, e.g., values below and above 20-40°C, 40-95%, -40-40 ($\mu$ mol m$^{-2}$s$^{-1}$), 0-1000 (Wm$^{-2}$) and 0-3000 ($\mu$ mol m$^{-2}$s$^{-1}$) for T$_{air}$, RH$_{air}$, $FCO_2$, SW$_i$, and PAR$_i$, respectively.

**24.** L305. What is the typical pattern of tropical forests?

**AC:** In the present study, the typical pattern of tropical forests refers/is limited to $FCO_2$ expected values during the day (e.g., maximum negative close to 10–12h (LT) and positive values starting around 18h (LT), usually lower than 5-10 $\mu$

mol m$^{-2}$s$^{-1}$ (nighttime) and 40 $\mu$ mol m$^{-2}$s$^{-1}$ (daytime), respectively (Vourlitis et al., 2011, von Randow et a., 2004, Gu et al., 2001). We have inserted a few sentences about it in the manuscript. We mention the typical pattern in the text to clarify the sentence.

**25.** L326. Statistical difference of what, exactly?

**AC:** Statistical (descriptive) parameters such as mean, median, coefficient of determination (R$^2$), and standard deviation present similar values. We mention these parameters in the text to clarify the sentence.

**26.** Figures 4–9. The authors often refer to top panel/bottom panel of these figures, but they are mostly side by side. I suggest labelling them with (a) and (b) and edit text accordingly. Also, in many captions, the authors could describe the figures in a bit more detail, and avoid using "correlation" as a synonym of "scatter plots".

**AC:** Ok, thank you very much for the suggestions. As mentioned, we labeled several figures with (a) and (b) and edited the text accordingly. Moreover, we've checked and improved the captions of some figures, describing them in a bit more detail. We checked both terms "correlation" and "scatter plots" and adjusted the text accordingly.

**27.** Figure 5. The authors present the binned averages as points, but presumably each bin has a significant variability that should be acknowledged/quantified.

**AC:** We agreed! Each bin has a significant variability. We determined and quantified the variability in terms of standard deviation (STD) for each bin and put them in the captions of Figures 5a and 5b. Moreover, additional figures are presented in the supplementary material with the STD (Figures S12a and S12b).

**28.** Table 4. Last header column should be statistic.

**AC:** Ok, done. We have replaced it with "statistic".

**29.** L346-352. The discussion attributes the variation and increase in PARd to aerosol dispersion, but couldn't that be partially attributed to clouds too? Presumably the solar zenith angle colours could be telling something on the seasonality, but this is not discussed in the text. Likewise, Table 4 is not really discussed, and I wonder if this is needed in the main text.

**AC:** Yes, but in minor proportion, especially because our methods take advantage of AOD AERONET (LEV20), measured only under cloudless conditions. Thus, we developed an algorithm to get the clear sky solar irradiance considering long-term measures of AOD (LEV20) (as described in the 2.3.2 section). In these conditions, the parameter $f$ can be used as a sensible and suitable indicator of clouds and aerosols if it is normalized by the $S_0(t)$, calibrated for these different sky conditions. However, our analyses cannot assertively state the complete absence of clouds. Regarding the $SZA$ (colorbar) we have made a few sentences to discuss the seasonal variations and their effects on $PAR_d$. As for Table 4, we have removed it from the main text and inserted it in the support material (Table S4).

**30.** L376. Most of this paragraph discusses LUE but no direct link with the results of this study is provided.

**AC:** Ok, we agreed. We have improved the paragraph with sentences to report the direct link with the results of this study is provided.

**31.** L414. Alta Floresta (2 words)?

**AC:** Ok, we have checked and corrected it. It's the name of a municipality.

**32.** L415. Where do we see the standard deviation?

**AC:** The standard deviations can be found in the previous studies published by Oliveira et al. (2007) in the FLONA-Tapajós (PA), Cirino et al. (2014) in Manaus (AM), and Ji-Paraná (RO). We have cited these studies in the text.

**33.** L415-417. Either discuss what the readers should get from Table 5 or move it to Supplemental Materials.

**AC:** As for Table 5, we also have removed it from the main text and inserted it in the support material (Table S5). A few sentences about this table were inserted in the manuscript's main text.

**34.** L445. Consider replacing "jumped" with "declined"

**AC:** Ok, done.

**35.** L485-487. This sentence seems to contradict the text in L474-481, and the authors did not present a clear separation between cloud and aerosol effects on NEE. I suggest dropping the sentence.

**AC:** Ok, we agreed. We deleted the sentences (L485-487).

**36.** Data availability. The authors should consider depositing their code to a permanent archive too. I also found the Ameriflux remark unnecessary, considering that the authors provide a DOI link with the data (although I had to remove the .2 at the end to access it).

**AC:** Ok, we agreed. In the final document, we deleted the Ameriflux remark from the Data availability section. Moreover, we checked the hyperlink in Table S3 and noticed the DOI: m5h5fw872g/1, which is promptly accessed by dataset: Cirino, Glauber; Vourlitis, George; Silva, Simone; Palácios, Rafael (2023), "Brazil-FluxMet-Stf", Mendeley Data, V1, doi: 10.17632/m5h5fw872g.1.

**Referee Comments #2**

**General comments:**

This study uses a series of satellite and ground observation to investigate the role of biomass burning aerosols on surface radiation and net ecosystem exchange in the northern part of the Mato Grosso State in Brazil. The work addresses a very interesting topic and I think brings an important contribution to existing literature in this area. However, I believe it still requires some important revisions before it can be published.

**AC:** We thank you again for your comments and suggestions. All considerations (major and minor comments and technical comments) are valuable and will be adhered to in the final version of the revised manuscript.

**Major Comments - RC#2**

**1.** The main weakness of the study in its current form is the absence of a thorough evaluation of the methodology proposed. In particular, it is important to present and discuss the extent to which estimates of key variables (e.g. solar irradiances, relative irradiance, leaf canopy temperature) compare with other existing estimates.

**AC:** That's right. We agreed. We have dedicated a brief period at the end of the main results sections to discuss how comparable the estimated key variables are with the methods/results used by other authors.

**2.** It is confusing in places that all the observed NEE changes are being entirely attributed to the aerosol-driven changes in diffuse radiation, while in fact they are the combined effect of several other additional key changes (e.g. temperature). Section 3.6 mentions this a bit, but in several instances in the manuscript this seems to be overlooked. Ideally it would be good to extend the methodology in order to allow isolating the temperature effect as well; however, if this proves to be very difficult, the current limitations and their implications should be clearly mentioned to avoid any confusions.

**AC:** It is a good point raised. (1) We can use atmospheric radiative transfer models (RTMs) coupled to the DVMs (Dynamic Vegetation Models), or (2) use our fits to try to separate and estimate the suggested effects. It seems more practical and doable to use our fit curves as long as 1–2 weeks are granted to us by the editorial. Anyway, we've improved our discussions as suggested. We added a brief paragraph to discuss aerosols' multiple effects on the biophysical variables that modulate photosynthesis rates.

**Minor Comments - RC#2**

**1.** I find the statements at lines 50-52 and 57-59 incorrect, as existing studies (including some already cited here) have in fact looked into this and quantified the impact of fire aerosols on plant CO2 absorptions.

**AC:** We have checked the sentences, corrected/replaced them with the sentences below, and reinforced the novelty of the present study.

"However, little research has been done on the ecotones in the Amazon, e.g., in the Cerrado-Amazonian Forest transition, which lies within the arc of deforestation, and other biomes such as Cerrado-Caatinga, Cerrado-Atlantic Forests, and Pantanal forests. Numeric simulations have also demonstrated the impact of aerosols on $GPP$ on the regional (Moreira et al., 2013; Rap et al., 2015; Bian et al., 2021) and global scales (Mercado et al., 2009; Rap et al., 2018), but they did still not pay enough attention to the physical representations of these impacts on these transition ecosystems, take into account their specificities. To our knowledge, this is the first study with this purpose."

**2.** Section 3.1: Should explain a bit more where do the differences in NEE values compared other estimates come from? You list some of the differences compared to the values from Vourlitis et al. (2011), but should also add a discussion on reasons and implications of those differences.

**AC:** Okay, done! We have already made this clear in the manuscript (revised). The difference between the values presented by Vourlitis et al. (2011) and the results of this research consists of the difference between the scientific questions and the underlying methodological approaches. E.g., most of the results presented by Vourlitis et al., are on a daily or monthly basis, while this research uses an approach hourly in fragments day specifics.

**3.** Could you please clarify if f, as defined in Section 2.3.3. can take values larger than 1 and what do they correspond to? These seem to be mentioned within the text (e.g. Sections 3.3 and 3.5).

**AC:** This is an excellent point. Yes, $f$ can take values larger than 1, usually due to the called "cloud gap effect" (Gu et al., 1999; 2001). In general, there are multiple scatterings of solar radiation by the clouds around the study area, but still out of the pyranometer's viewing angle. However, there is still no consensus about the amounts in the literature. It has been observed $f$ of about 1.1-1.3 for the southern Amazon (Gu et al., 2001). We have added some sentences about this effect to clarify the results presented in Figure 8a.

**4.** Throughout the manuscript, there seems to be an assumption that biomass burning is the only aeosol species affecting surface radiation and NEE. It would be good to discuss the extent to which other aerosol species (e.g. different aerosol optical properties (e.g.single scattering albedo) affect radiation and NEE?

**AC:** We agreed! We have inserted a few sentences to clarify this discussion. BBOA is the predominant aerosol in the region, especially during the burning season, but about of 10% of the burn plume load is composed of BC (Black Carbon) and BCr (Brown Carbon), for which the Single Scattering Albedo (SSA) is affected. In general, these particles have the potential to heat the atmosphere (absorption is greater than reflection), producing values that may be above the optimal physiological thresholds of the ecosystem influencing the $NEE$.

**Technical Corrections - RC#2**

**1.** Line 50: "litter" should be "little".

**AC:** Done! We replaced it with 'little'.

**2.** Line 76: What is meant by "region of 105 continuous agricultural expansion"?

**AC:** Ok, done. It was a typing error.

**3.** Fig 4.: Should clarify in caption what are the NEE values illustrated based on.

**AC:** We hope this has clarified what the $NEE$ values illustrated. The revised caption is below:

$NEE$ average hourly cycle between June/2005 and July/2008, during the rainy (a) and less rainy (b) for the semideciduous forest at the Claudia municipality, in the local study (results from the eddy-covariance system). No filters are applied. The $NEE$ is presented for any sky conditions during the year. The standard deviation is shown as vertical bars.

**4.** 5-8: Unclear what is meant by top and bottom panels (mentioned both in the figure captions and within the text).

**AC:** We have done. We removed all observed cases and replaced them with single letters (e.g., a, b, c, …) as per figure citations in the manuscript.

**5.** Line 502: "Relative irradiance" instead of "Irradiance".

**AC:** Ok, done.

**6.** There are several grammar/syntax errors throughout the text that need to be corrected.

**AC:** Please accept our apologies for grammatical errors. We have made some corrections throughout the manuscript and hope to have improved the quality of the written text. Anyhow, we're still going to revise and take a look carefully several times.

**7.** Should revise the incorrect use of ";" throughout the manuscript.

**AC:** We checked for inappropriate use of ";" throughout the text. The incorrect cases were deleted.

**Additional References**

Alencar, A., D. Nepstad, D. McGrath, P. Moutinho, P. Pacheco, M.C. Vera Diaz & B. Soares Filho. 2004. Desmatamento na Amazônia: indo além da emergência crônica. Instituto de Pesquisas Ambientais da Amazônia (IPAM), Belém, Brasil.

Anderegg, W.R.L., Trugman, A.T., Badgley, G. et al. Divergent forest sensitivity to repeated extreme droughts. Nat. Clim. Chang. 10, 1091–1095 (2020). https://doi.org/10.1038/s41558-020-00919-1

Andrade, A. M. D. et al Downward longwave radiation estimates for clear and all sky conditions in Central Amazonia. Revista Brasileira de Climatologia, ano 17, vol. 28, ´p. 602-618, JAN/JUN 2021.

Bird, R. E. and Hulstrom, R. L. (1981b) A simplified, clear sky model for direct and diffuse insolation on horizontal surfaces, Technical Report SERI/TR-642-761, Solar Research Institute, Golden, Colorado.

Brando, P.M., Balch, J.K., Nepstad, D.C., Morton, D.C., Putz, F.E., Coe, M.T. et al. (2014) Abrupt increases in Amazonian tree mortality due to drought-fire interactions. Proceedings of the National Academy of Sciences, 111, 6347– 6352.

Brando, P.M.; Silvério, D.; Maracahipes-Santos, L.; Oliveira-Santos, C.; Levick, S.R.; Coe, M.T.; Migliavacca, M.; Balch, J.K.; Macedo, M.N.; Nepstad, D.C.; et al. Prolonged tropical forest degradation due to compounding disturbances: Implications for CO2 and H2O fluxes. Glob. Chang. Biol. 2019, 25, 2855–2868.

Corripio, J. G.: 2003, Vectorial algebra algorithms for calculating terrain parameters from DEMs and the position of the sun for solar radiation modelling in mountainous terrain, International Journal of Geographical Information Science 17(1), 1-23.

Davison, C. W., Rahbek, C., & Morueta-Holme, N. (2021). Land-use change and biodiversity: Challenges for assembling evidence on the greatest threat to nature. Global Change Biology, 27(21), 5414–5429. https://doi.org/10.1111/gcb.15846.

Durand, M., Murchie, E. H., Lindfors, A. V., Urban, O., Aphalo, P. J., & Robson, T. M. (2021). Diffuse solar radiation and canopy photosynthesis in a changing environment. Agricultural and Forest Meteorology. https://doi.org/10.1016/j.agrformet.2021.108684

Gonsamo, A., & Chen, J. M. (2018). Vegetation Primary Productivity. Comprehensive Remote Sensing, 163–189. doi:10.1016/b978-0-12-409548-9.10535-4.

Gu L, Baldocchi DD, Wofsy SC, Munger JW, Michalsky JJ, Urbanski SP, Boden TA. Response of a deciduous forest to the Mount Pinatubo eruption: enhanced photosynthesis. Science. 2003 Mar 28;299(5615):2035-8. doi: 10.1126/science.1078366. PMID: 12663919.

Heloisa Oliveira Marques, Marcelo Sacardi Biudes, Vagner Marques Pavão, Nadja Gomes Machado, Carlos Alexandre Santos Querino, Victor Hugo de Morais Danelichen, "Estimated net radiation in an Amazon–Cerrado transition forest by Landsat 5 TM," J. Appl. Rem. Sens. 11(4) 046020 (14 December 2017) https://doi.org/10.1117/1.JRS.11.046020

Hubau, W., Lewis, S.L., Phillips, O.L. et al. Asynchronous carbon sink saturation in African and Amazonian tropical forests. Nature 579, 80–87 (2020). https://doi.org/10.1038/s41586-020-2035-0

Idso, S. B., & Jackson, R. D. (1969). Thermal radiation from the atmosphere. Journal of Geophysical Research, 74(23), 5397–5403. doi:10.1029/jc074i023p05397.

Monteith JL, Unsworth MH. 1990. Principles of environmental physics ( 2 nd edn). London, UK: Edward Arnold.

Ometto, J.P., K. Kalaba, G.Z. Anshari, N. Chacón, A. Farrell, S.A. Halim, H. Neufeldt, and R. Sukumar, 2022: Cross-Chapter Paper 7: Tropical Forests. In: Climate Change 2022: Impacts, Adaptation and Vulnerability. Contribution of Working Group II to the Sixth Assessment Report of the Intergovernmental Panel on Climate Change [H.-O. Pörtner,

D.C. Roberts, M. Tignor, E.S. Poloczanska, K. Mintenbeck, A. Alegría, M. Craig, S. Langsdorf, S. Löschke, V. Möller, A. Okem, B. Rama (eds.)]. Cambridge University Press, Cambridge, UK and New York, NY, USA, pp. 2369–2410, doi:10.1017/9781009325844.024.

Reboita, M.S., Krusche, N., Ambrizzi, T., da Rocha, R.P., 2012. Entendendo o Tempo e o Clima na América do Sul. Revista Terra e Didática, 8(1), 34-50. Disponível: https://doi.org/ 10.20396/ td.v8i1.8637425.

Saatchi, S.; Longo, M.; Xu, L.; Yang, Y.; Abe, H.; Andre, M.; Aukema, J.E.; Carvalhais, N.; Cadillo-Quiroz, H.; Cerbu, G.A.; et al. Detecting vulnerability of humid tropical forests to multiple stressors. One Earth 2021, 4, 988–1003.

Silva Junior et al., 2020 C.H.L. Silva Junior, L.E.O.C. Aragão, L.O. Anderson, M.G. Fonseca, Y.E. Shimabukuro, C. Vancutsem, et al. Persistent collapse of biomass in Amazonian forest edges following deforestation leads to unaccounted carbon losses Sci. Adv., 6 (40) (2020)

Sullivan, M. J. P. et al. Long-term thermal sensitivity of earth's tropical forests. Science 368, 869–874 (2020).

Venturini, A. M., Gontijo, J. B., Mandro, J. A., Berenguer, E., Peay, K. G., Tsai, S. M., & Bohannan, B. J. (2023). Soil microbes under threat in the Amazon Rainforest. Trends in Ecology & Evolution.

Von Randow, R. C. S.; Alvala, R. C. S. Estimativa da radiação de onda longa atmosférica no Pantanal Sul Mato-Grossense durante os períodos secos de 1999 e 2000. Revista Brasileira de Meteorologia, v. 21, n. 3b, p. 398-412, 2006.

[Figure]

Figure 1: shows (a) the hourly cycle of leaf canopy temperature estimated by the $LC_T$ model, Equation 9 (Tribusy, 2005); (b) the hourly cycle of air temperature; and (c) the difference found between $LC_T$ and $T_{air}$ (°C), indicating the warmest leaf canopy than the air between 07–17h (LT).

[Figure]

Figure 2: shows (a) the hourly cycle of leaf canopy temperature estimated from Equation 1; (b) the hourly cycle of air temperature; and (c) the difference found between $LC_{Ts}$ and $T_{air}$ (°C), illustrating the warmest leaf canopy during daylight, between 07–17h (LT).

[Figure]

Figure 3: shows the monthly changes on the $NEE_0(sza)$. Fit curves (Table 1) under clear sky conditions ($f \sim 1.0$) as a function of the $SZA$ ranges, between Jun2005-Jul2008, 07–17h (LT). The black dot-line is the $NEE_0(sza)$ average used in the preprint version (Figure 3). This figure will replace Figure 3.

[Figure]

Figure 4: shows hourly cycle changes on the $NEE_0(sza)$. Fit curves under clear sky conditions ($f \sim 1.0$) for each month of the year between Jul/2005-Jun/2008, 07-17h (LT).

[Figure]

Figure 5: shows the %NEE calculated from Equation 11 without correction on the $NEE_0(sza)$ of Figure 8b (preprint-version), between Jul/2005-Jun/2008. This figure will be replaced with Figure 6, below.

[Figure]

Figure 6: shows %NEE calculated from Equation 11 (preprint-version), corrected with the $NEE_0(sza)$ computed from the fit curves presented in Table 1, between Jul/2005-Jun/2008.

Table 1: coefficients of the $NEE_0(asz)$ curves adjusted monthly to clear sky conditions in the municipality of Claudia-MT between Jun/2005 and Jul/2008.

| Months | p1 | p2 | p3 | $R^2$ | RMSE |
|---|---|---|---|---|---|
| January | +0.0014 | +0.0826 | −13.90 | 0.6 | 4.6 |
| February | +0.0054 | −0.3876 | −04.60 | 0.4 | 5.6 |
| March | +0.0011 | +0.1145 | −14.10 | 0.3 | 5.4 |
| April | +0.0033 | −0.0370 | −13.50 | 0.5 | 5.8 |
| May | +0.0057 | −0.2150 | −10.20 | 0.4 | 4.4 |
| June | +0.0091 | −0.6277 | −01.00 | 0.2 | 7.2 |
| July | +0.0039 | −0.1793 | −08.50 | 0.1 | 3.4 |
| August | +0.0058 | −0.2657 | −08.60 | 0.4 | 5.3 |
| September | +0.0018 | +0.0045 | −11.40 | 0.3 | 4.8 |
| October | +0.0015 | +0.0762 | −14.10 | 0.4 | 5.4 |
| November | +0.0026 | +0.0527 | −14.80 | 0.6 | 5.0 |
| December | +0.0117 | −0.7177 | −03.40 | 0.8 | 4.0 |

---

## Author Response (AR1)

**Final Response**
**egusphere-2023-684**
**Rodrigues et. al (2023)**
**simone.silva@ig.ufpa.br**

[Figure]

Responses and actions on referee comments on the manuscript egusphere-2023-684, Enhanced net $CO_2$ exchange of a semi-deciduous forest in the southern Amazon due to diffuse radiation from biomass burning, by Simone Rodrigues et al., 2023.

Date: November 7, 2023

**Authors' General Comments:**

We are grateful to both reviewers for all their comments on our manuscript. We have carefully reviewed and complied with all suggestions/corrections mentioned by reviewers. Furthermore, we brought new results to address and discuss the potential confounding effects (Figs. 3, 8a, 8b, S8, S11, and S13). We seek to evaluate our assumptions clearly and better support them (Figs. S5, S6, TabS4), as pointed out. Our answers are just below each comment, tracked in blue color. To facilitate the review work, we hyperlinked below all the references and results (revised and expanded) and included them in the supplementary material. We hope to have improved the manuscript, but we are also willing to insert new analyses if necessary to improve the quality of the article further.

We also want to highlight two relevant points of this research (novelty and benefits): — (1) our most important scientific issue refers to the responses of transition forests to the aerosols and their impacts on the carbon uptake capacity at the ecotone Cerrado-Amazon Forest. As far as we know, this is the first study to report these effects in the transition between the Amazon and Cerrado biomes, which are considered critical in central Brazil. The novelty was analyzing the impact of (BBOA) emissions on the $NEE$ in a semideciduous forest ecosystem in the southern Amazon basin from the local measurements (in-situ). The manuscript highlights the inflection points (critical points) from which the photosynthetic process drastically decays in the studied ecotone. We also developed a clear-sky solar irradiance algorithm based on the long-term measurements from the AERONET system, adjusted to the environmental conditions in the region concerning the vertical profile of trace gases (attenuators). However, while it is a useful by-product, it was not our focus. — (2) our results can improve Dynamic Vegetation Modelling (DVMs) in these ecotones. In addition, our study provides useful information for global public policies for preserving pan-tropical ecosystems in the face of the impacts caused by climate change, intensified by no-climate factors, such as anthropogenic actions.

**Referee Comments #1**

**General comments:**

The carbon cycle in the Amazon has been directly and indirectly impacted by both climate change and land use/land cover change, but the effect of each driver on net ecosystem exchange (NEE) remains uncertain. In this study, Rodrigues, Cirino and colleagues seek to use a combination of observations from an eddy covariance tower, and aerosol optical depth estimates from in situ and satellites to assess the impact of biomass burning aerosols on the radiation partition (direct/diffuse) and how it cascades into the carbon fluxes in a semi-deciduous forest site in the southern part of Amazon.

The topic is well suited and relevant for Biogeosciences, and the authors have data and analytical tools to provide an important contribution. However, the current analyses have important assumptions that are not clearly evaluated, and potential confounding effects are not addressed or discussed. In addition, at many places in the Results and Discussion section, the authors make statements that are not clearly supported by the results. I list some of these points below. Most of these concerns are fixable, but it will likely require substantial revision of the methods, analyses and discussion.

Most of the analyses presented here assumes that the variability of NEE is driven by gross primary productivity (GPP), and most of the variability in light quality is driven by aerosols. Ecosystem respiration, which had been pointed out previously as the main driver of seasonal variation of NEE in the same area (Vourlitis et al. 2011) is not mentioned as a potential confounding effect. More generally, if the goal of the research is to assess the effect of smoke on diffuse light and NEE, I wonder why the authors did not restrict the analysis to the dry seasons of the study period. That would likely reduce confounding effects due to seasonality (e.g., water stress, deciduousness, ecosystem respiration) and potentially provide better support for most of the assumptions in the derived quantities described in the methods.

**AC:** We thank the referee for pointing out this fundamental issue. Indeed, $NEE$ could be influenced by the ecosystem respiration (seasonally dependent). We have corrected the analysis by including a monthly-based $NEE$ fitting, which results, on average, in a small variation in the final outcome (see answer to specific points below). As for the unsupported claims, we generated additional results to support them and discussed the results in the revised manuscript (Figs. S5, S6, TabS4), as already mentioned above. We have extended the supporting material by adding the procedure to calculate the $GPP$ (Fig. S11). Regarding potential confounding effects, additional results/discussions have been inserted (see the answer below – "Specific Points"). The fit between $NEE$ and $SZA$ (see Figures 3 and S8) has been estimated for each month of the year along the dataset to remove or minimize the effects due to seasonality, as suggested. From these corrections, it observed an average increase of about 15% in the %$NEE$ to $SZA$ 0-75° but significant statistically around 45% to $SZA$ 50-75° (Figure 8a-b). The revisions and modifications indicated that the $NEE_0(sza)$ must be adjusted as a function of seasonally to avoid sub-estimates of the effects of aerosols on photosynthetic rates. We also want to highlight that the results reached from these suggestions enriched the quality of the manuscript.

**Specific points - RC#1**

**1**. Leaf canopy temperature (Section 2.3.7). The model used to estimate this quantity seems to come from Tribuzy et al. (2005), and applied to the focus site in Mato Grosso. Interestingly, Equation 9 does not depend on air temperature. Perhaps this is less of an issue for Manaus, the equatorial site where this equation was originally developed. However, as the authors indicate, Mato Grosso does experience temperature variations from weather systems. Moreover, in Figure 9, the air temperature range is broader than canopy temperature, which suggests that the modelled canopy temperature fails to capture the actual variability. I understand the authors do not have any validation data, but they should consider this limitation when using the canopy temperature estimates".

**AC:** In the revised version of the manuscript, we have clarified the limitations of the $LC_T$ model. We agree that $T_{air}$ is broader than $LC_T$. However, $T_{air}$ is systematically smaller throughout the day (Figure 1a and Figure 1b) below, results obtained from the same data points shown in Figure 9 (preprint version). On average, the amplitude between $LC_T$ and $T_{air}$ is equal to 2.2 ($\pm$ 2.1) °C, and the leaf canopy is warmer than the air between 7-17h (LT), as expected (Figure 1c).

However, to verify the issue appointed and evaluate the consistency of the $LC_T$ model (Tribuzy, 2005), a second method was tested ($LC_{Ts}$) based on the Stefan-Boltzmann equation (Doughty et al., 2010; Cirino et al., 2014). We observed similar behaviors compared to Figure 1, i.e., leaf canopy warmer than $T_{air}$ during the day (Figure 2a and Figure 2b). However, the $LC_{Ts}$ standard deviation (std) is significantly higher. On average, the amplitude between $LC_{Ts}$ and $T_{air}$ is about 1.7 ($\pm$ 11.1) °C (Figure 2c). $LC_{Ts}$ seems to capture the average hourly behavior but gives a grossly greater variability hourly compared to $T_{air}$ (7-17h, LT) during the year. Both Figure 1 and Figure 2 are inserted in the revised supplementary material as Figures S9 and S10. From our perspective, the model $LC_T$ provides an acceptable average hourly pattern for leaf canopy temperature, although the std is unrealistic compared to $T_{air}$, which is broader than canopy temperature. Due to the $LC_T$ limitations, we are probably underestimating the uncertainties of aerosol effects on canopy temperature. However, the impact of aerosols on $Tair$ also indicates an important cooling at the surface ($\sim$ 3-4 °C), with relevant effects on the canopy and for the functioning of the ecosystem studied. Therefore, we've kept Figure 9 (preprint version) in the body of the manuscript and briefly discussed the limitation when using the estimates $LC_T$, as suggested. We inserted $LC_{Ts}$ the model below (Equation 1) in the support material, as Equation S1.

$$LC_{Ts} = \sqrt[4]{\left( \frac{SWi(1 - \alpha + LWa - Rn)}{\varepsilon\sigma} \right)} \tag{1}$$

Where: Rn is the net radiation measured in the experimental area (Wm$^{-2}$); $\alpha$ is the mean albedo of the leaf canopy during the dry and wet seasons, respectively equal to 0.079 and 0.126 [Marques et al., 2017]; $LW_a$ is the atmospheric longwave radiation in Wm$^{-2}$ [Idso and Jackson, 1969], adjusted for the polluted atmospheric conditions of Mato Grosso during the dry season [Von Randow, 2006]; $\epsilon$ is the emissivity, assumed to be 0.98 [Monteith and Unsworth, 1990, Marques et al., 2017] and, $\sigma$, the Stefan-Boltzmann constant ($5.67 \times 10^{-8}$ Wm$^{-2}$ K$^{-4}$). Similar methods have been widely used in the absence of direct measurements of leaf temperature in the Amazon (Doughthy et al., 2010; Cirino et al., 2014; Aguiar et al., 2012; [Andrade et al., 2021]). The discussions inherent to canopy temperature estimates are in lines (L79-L81 and L482-L495).

**2**. Clear-sky NEE (Section 2.3.8). I wonder about what this equation is actually capturing. The same SZA may mean different times of day (and temperature and VPD) at different times of the year, so water and heat stress and deciduousness may be adding time-dependent noise on the fitted model (Eq. 10). Perhaps the authors should include month as a factor in their models? Or eliminate seasonal confounding effects by focussing on the dry season only?

**AC:** This is a good point, and we fully agree. To remove confounding effects, we recalculate $NEE_0(sza)$ using a curve for each month of the year Figure 3. Curve coefficients are listed in Table 1. Figure 4 shows the changes on the $NEE_0(sza)$ during the year, which suggests corrections in the %$NEE$ calculation (Figure 8b, manuscript). As mentioned before, we observed an average increase of about 15% ($SZA$ 0-75°) and 45% ($SZA$ 50-75°) in %$NEE$ after corrections were applied (see Figure 5 and Figure 6). Based on these results, we updated Figs. 3 and 8b of the manuscript. This procedure minimized and corrected the potential confounding effects of $NEE$ seasonality. Thus, we decided to keep the rainy and transition months in our analyses between 2005-2008, considering the following reasons: (1) the relative contribution of local fires during the wet season is small, about $\sim$10% of the sample space, but it still contributes to improving the statistics, which is a critical aspect of the study; (2) despite the isolated fires in the study region (wet season), strong contribution of regional and long-range transport (Holanda et al., 2023) for biomass burning aerosols is expected (Figs. S4, S5, and S6, supp. material); (3) removing or keeping transitional and rainy periods in the analyses, does not change the scientific direction of the results originally found in Figure 8b and the underlying conclusions (see Figure 5 and Figure 6). Please track changes and the updated subsection (3.5.1) with a brief discussion of the potential confounding factors (L297-303, L438-444).

**3**. Results and discussion section. I found this section difficult to follow, and often found myself unsure on whether the authors were describing their results or previous research. If the authors prefer to keep the results and discussion together (as opposed to separate sections), I suggest reorganising the paragraphs so the distinction between results from this research and the discussion within the broader literature is very clear. For example, in Section 3.1, it was difficult to separate which results were from this study and which results came from Vourlitis et al. (2011). Perhaps focus more on describing Figure 4 and only briefly compare/contrast with the previous results by Vourlitis et al. (2011).

**AC:** We have reorganized the paragraphs, giving greater attention to our results and briefly discussing the similarities in the

broader literature. To better organize the "Results and discussion" section, we followed (whenever possible) the following order: (1) a description of our results and (2) a discussion thereof (i.e., a brief comparison of relevant aspects with results of previous studies). Likewise, we updated the paragraph of Figure 4 according to the suggestion. Lines (e.g., L339-354 and L355-373).

**4**. In addition, the authors used NEE and GPP interchangeably throughout most of the manuscript (including in the definition of light use efficiency). Within the same season and during daytime, I can see that this is less likely a problem, but unless I missed this, the authors used the time series across all seasons, which could confounding effects. I understand that GPP estimates from eddy covariance flux towers can be uncertain too, but I think the authors could explain why they opted for analysing NEE instead of GPP. Also, the authors highlight that the forest is semi-deciduous, which made me wonder about the mechanisms that could lead to an increase in NEE through increase in GPP during the dry and smoky season. The assumption of GPP-driven variability in NEE appears multiple times (e.g., L439-L457, L.494-502), so it seems central to the discussion, yet it is not fully supported by the presented data and analyses.

**AC:** Since no direct local measurements of Ecosystemic Respiration ($R_{eco}$) exist, estimates are necessary. However, typically, the models available in the literature grossly overestimate or underestimate the local $R_{eco}$, especially when in-situ data are unavailable to fit them (e.g., autotrophic and heterotrophic respiration; litter, soil, trunks, branches, leaves, and roots) [Malhi, 2012]. It is also important to highlight that Vourlitis et al. (2011) made daily estimates for the $R_{eco}$ without isolating the effects of pollution from fires. These conditions are pretty different for the current study. Furthermore, as this kind of study was unexploited in the "Cerrado-Amazon Forest" ecotone, we still do not know how aerosols modify the region's $R_{eco}$. Once photosynthetic rates are also modulated by solar radiation (attenuated by BBOA), changes in the VPD may also be impacting ecosystem respiration from no-linear interactions, influencing the opening and closing of stomata, canopy temperature, humidity, and soil temperature. All these factors influence the microbiological dynamics of the soil and litter, with implications still unknown for $R_{eco}$ (in-situ). Thus, we assume that the uncertainties underlying the calculation of the $R_{eco}$ (for the reasons mentioned) could affect our results equally significantly by disregarding it. Therefore, we assumed $GPP \approx NEE$ on a monthly basis. (L495-521)

We agreed that Deciduousness and flush new leaves are seasonal phenomena independent of aerosol concentration (BBOA). However, both occur sequentially during the burning season, between Jul-Nov, at the site [IBGE, 1992] (Fig. S3). To better support our assumption ($GPP \approx NEE$), we have done a similar test (Figs. 8b and S12) using only daytime data during the dry season (Figs. S12 and S13). We found that the % change (on average) is similar, around 15% for $SZA$ (0-75º) and around 35% for $SZA$ (50-75º). Both differences may be attributed to uncertainties due to the daytime $R_{eco}$ (Figure S11c) and factors that need to be better explored in future work. We hypothesized some mechanisms that could lead to an increase in $NEE$ throughout the dry and smoky seasons (strongly supported by previous studies in the Amazon and world), as follows below. A brief discussion in the 3.6 section was added (L495-521).

[1]. During the dry season (Jul-Sep), photosynthetic deficit due to deciduality is partially compensated by positive feedback of extrinsic factors: BBOA concentration, PAR(D), cooling of the air and leaf canopy, and VPD reduction [Vourlitis et al., 2001, Gu et al., 2003, Rap et al., 2018, Corwin et al., 2022].

[2]. During the flush new leaves season (Oct-May), photosynthetic enhancement is primarily explained by positive feedback both extrinsic (mentioned) and intrinsic factors (Leaf Area Index and LUE): variation in the characteristics of the forest canopy due to the newly sprouted leaves, i.e., higher photosynthetic capacity of canopy that compensates the unfavorable stomatal response due to precedent drought [Wu et al., 2016, Green et al., 2020];

[3]. The magnitude of the effects observed in assumptions (1) and (2) will strongly depend on the intensity and frequency of occurrence of meteorological phenomena (e.g., planetary limited layer dynamics) [Fuentes et al., 2016, Gao, 2020];

**5.** Likewise, the authors discuss the effects of potential confounding effects on the observed relationship between diffuse light and NEE in Section 3.6 and in Figure 9 (e.g., temperature and vapour pressure deficit), but they do not account for these other important drivers in their analyses. They mention in line 508-509 that they could not quantify the effect of these variables in this study, but they do not explain why, and I do not see any reason for not exploring it with statistical models that account for these other variables (similar to the models that they implemented, but with additional predictors).

The authors did attempt to mitigate these effects by exploring the response of NEE to diffuse radiation fraction by binning data by solar zenith angle (Section 3.5, Figure 8), but that may have caused the bins to have different times of day grouped together across seasons, so it is difficult to interpret the results.

**AC:** It is a good point raised. We have two paths to follow: (1) atmospheric and canopy radiative transfer models coupled (e.g., Dynamic Vegetation Models – DVMs), or (2) our fits to try to separate and estimate the suggested effects. From our standpoint, this is outside the scope of this study and will be investigated in future studies. So, we added some sentences to discuss/mention the complex and non-linear relationship between photosynthesis, water vapor, and leaf canopy temperature variables that modulate photosynthesis rates (L545-558). As for the bins having different times of day, note that it is a complex trade-off between opposing factors: we took the smallest possible intervals to avoid interpretation difficulties and, at the same time, conserve sample space. To minimize these effects, our results are analyzed by several SZA ranges.

**Minor points - RC#1**

I found the text to have several typographic mistakes and sentences that appear out of place. I am not listing every one, but I suggest the authors to thoroughly revise and streamline the revised text for clarity.

**AC:** Thank you for reviewing the article in detail. We've reviewed the notes highlighted below and corrected all points noted. In addition, we thoroughly revised the article for gross typographical errors and misplaced sentences, as suggested.

**1.** L1. The opening sentence is a bit circular (atmospheric processes and climate are closely linked to carbon cycle as a consequence of biosphere-atmosphere coupling).

**AC:** We have replaced the sentence mentioning that carbon cycling in the Amazon fundamentally depends on the functioning of ecosystems and atmospheric dynamics, which are highly intricate **(L1-2)**.

**2.** L2. The radiative effects of aerosols and clouds ON XYZ are still unknown…

**AC:** We have replaced the sentence with the following one: "The radiative effects of aerosols on the Amazon-Cerrado biomes still need to be discovered for a wide variety of vegetation, usually unconsidered since that was not inventoried in previous studies **(L2-L3)**."

**3.** L5. Relative irradiance: briefly explain "relative" (i.e., relative to which conditions)

**AC:** We agree and have rewritten it as follows: "Our results show a decrease in incident solar radiation of up to 40% in regard to smoky sky conditions (i.e., a decrease in relative irradiance $f$ from 1.10 to 0.67). Consequently, we observed an average increase in the carbon uptake ($\%NEE$) of 20%, 40%, and 110% for $SZA$ ranging between 0-25°, 25-50°, and 50-75° (deg), respectively **(L6-7)**."

**4.** L10. 10% increase or 10% decrease?

**AC:** We have revised and rewritten it: "… Important influences on VPD and air temperature and canopy induced by the interaction between solar radiation and high aerosol load in the observation area were also noticed. On average, an increase up to 2-3 hPa and a cooling of about 3-4 °C is observed, respectively…" **(L9-12)**

**5.** L17-20. The nature of the debate is unclear, consider briefly explaining it.

**AC:** We have added sentences to explain the nature of the debate, and we made it clearer as following: "...The role of Amazonian Forest ecosystems has been widely debated in the context of global climate change. Redistribution of biomes and plant species [Davison et al., 2021], loss of biodiversity [Brando et al., 2014, Saatchi et al., 2021], increase in fires [Brando et al., 2019, Alencar et al., 2022, Sullivan et al., 2020], outbreaks of pests and diseases [Anderegg et al., 2020] are examples of impacts, aggravated not only by climatic factors but also by anthropogenic ones [Ometto and Kalaba, 2022].

These impacts have been threatening the largest pantropical $CO_2$ sinks since 1990'. Reductions from 1.26 PgC yr$^{-1}$ to 0.29 PgC yr$^{-1}$ are expected between 1990-2030, possibly reaching zero in the Amazon [Hubau et al., 2020] **(L16-23)**..."

**6.** L24-25. Mention other sources of CO2 too? Deforestation and degradation (including fires).

**AC:** Thank you for the suggestion. We have inserted other sources in the body text as follows: "...$CO_2$ absorption through photosynthesis increases the vegetation and soil C stocks, representing a C sink, while plants, animals, microbial respiration, decomposition of dead vegetal biomass, and wildfires release $CO_2$, representing a C source to the atmosphere (Artaxo et al., 2022; [Venturini et al., 2023, Silva Junior et al., 2020],)..." **(L26-29)**.

**7.** L36-39. I found this discussion somewhat misleading due to the significant difference in scale across the studies (Gatti et al. is a regional study, whereas the other references are for specific sites).

**AC:** We have rewritten the sentences as follows: "...However, regional numeric modeling (Moreira et al., 2017) and in-situ studies (Carswell et al., 2002; von Randow et al., 2004) indicate that Amazonian forests may occasionally be net sinks of atmospheric $CO_2$; or approximately at equilibrium (Vourlitis et al., 2011)..." (L39-41)

**8.** L55. The sentence is vague. What are the current limitations are why do these limitations matter?

**AC:** We agree. We reinforced the text with info about the current limitations and its matter, as follows: "...The models, however, need improvements in parameterizing the radiative effects of aerosols and clouds on the $NEE$, e.g., a more realistic representation of the canopy structure and processes leaf physiological and morphological [Durand et al., 2021]. Improvements in the aerosol optical model, its properties, secondary formation, lifetime, evolution, and absorption of aerosols are also critical [Drugé et al., 2022], especially those related to shape, size, and chemical composition. These improvements are fundamental for a more accurate and realistic spatial distribution of the atmospheric $CO_2$ absorption potential by Amazonian forests (Procópio et al., 2014; Moreira et al., 2017)..."(L-58-63)

**9.** L57-59. Doesn't Rap et al. (2015), which the authors already cite, discuss the effects of aerosols on productivity across the Amazon (including Mato Grosso) using numerical modelling?

**AC:** Rap et al. (2015; 2018) have studied the effects of aerosols on productivity across the Amazon Basin, considering only numeric simulations (DVGMs). The specifics of semideciduous forests have been unconsidered in both studies. (L56-57)

**10.** L76. Remove "105"? It seems out of place.

**AC:** Corrected.

**11.** L84–85. The areas presented in this sentence (49.95 km2 and 20.50km2) seem very small for Mato Grosso.

**AC:** We've corrected and updated these numbers and the citation as followed: "...The areas of transitional forests (Amazon Forest-Cerrado) covered approximately 41% (362,538 km²) of the State of Mato Grosso. Due to the advance of the agricultural frontier, 21% of these areas suffered drastic reductions. Part of these forest areas are found in protected areas and territories of indigenous communities (approximately 17%). The deciduous and semi-deciduous forests of the Cerrado biome initially covered 49,951 km² in the State of Mato Grosso. Deforested areas represented about ≈ 41% of this total, with only 14% located in conservation units in Alencar, etc. (2004). The geographic positions of these forests are discontinuous due to climatic fluctuations that have occurred in the last 10,000 years Prado and Gibbs (1993)..."(L91-97)

**12.** L94. Which systems operate in northern Amazon?

**AC:** We have mentioned the weather systems in the area of study as follows: "...This area's 30-year average annual temperature is 24°C, with precipitation of approximately 2000 mm yr$^{-1}$ (Vourlitis et al. 2002). The Bolivian High (BH) and South Atlantic Convergence Zone (SACZ) are among the active atmospheric systems in the northern Mato Grosso region, while the southern is affected by extratropical systems, such as Frontal Systems [Reboita et al., 2012]..." (L-102-104)

**13.** L97. Flush new leaves? "Recover" strikes me as the incorrect word, as deciduousness is an evolutionary adaptation to droughts.

**AC:** We have replaced the word as followed:"...The loss of leaves (deciduousness) during the dry season (July-September) is quite sensitive to water availability and temperatures (maximum and minimum) in the region. With the arrival of the wet season (November-May), flush new leaves occur with typical characteristics of tropical forests (Vourlitis et al. 2011)..."(L104-107)

**14.** Section 2.2.1. What is the time span of the AERONET data, 1993-2018 (L105) or 1993-2021 (L121).

**AC:** The data period has been corrected in the manuscript for 1999-2017.(L-114)

**15.** L143. Drop "in Amazonia" as eddy covariance has been used globally.

**AC:** We have replaced to: "...The eddy covariance system has been widely used to measure the net $CO_2$ flux by the ecosystem..."(L-151-152)

**16.** L148-149. Drop sentence? This does not seem to add much content.

**AC:** Done. That sentence was deleted.

**17.** L155-157. Sentence is confusing.

**AC:** We rewrite the sentences as followed: "...The $NEE$ is obtained from the eddy-covariance system. The eddy system provides $CO_2$ flux measurements at 10 Hz from a sonic anemometer (CSAT-3, Campbell Scientific, Inc., Logan, UT) integrated with an open-path gas analyzer (LI-7500, LI-COR Inc., Lincoln, NE). For $NEE$ calculation, the storage term $S[CO_2]$ is obtained according to Aubinet (2012) and Araujo (2010). For $S[CO_2]_p$ term calculation, we considered continuous measures of the CO2 concentration vertically arranged between the ground and the top of the tower (Vourlitis et al., 2011). Under these conditions, the $NEE$ of $CO_2$ is approximated by Equation 1..." (L161-166)

**18.** L170. Temperature should be in K, not °C, for equation 2.

**AC:** Thanks. We have verified and replaced °C with K, as noticed.(L178-179)

**19.** L189. What is Meteoexploration (SolarCalculator)? Provide reference/citation/context.

**AC:** We inserted three references and clarified the text. The updated sentences are as followed: "...The Solar-Calculator is a free system used to compute the clear sky solar irradiance, managed by Meteo Exploration company. The solar irradiance is calculated according to [Bird, R. E., Hulstrom, 1981], updated by [Corripio, 2003]. The hyperlinked for Solar-Calculator is listed in Table S5..."(L-204-206)

**20.** Equation 4. The notation is somewhat confusing. Perhaps replace the numerator with SWia(t), so it is universal (as opposed to only when AODa > 0.10 and accounting for cloud cover)?

**AC:** We change the numerator of Equation 4 as suggested and insert the typographical value $SW_i(t)$ throughout the manuscript. Moreover, we revised all typographical values in the text body. Equation 4, updated, is below: (L-226)

$$f = \frac{SW_i(t)}{S_0(t)\{AODa < 0.10, cloudless\}} \tag{2}$$

**21.** L244-245. The definition of LUE reads as a bit too circular to me.

**AC:** We have redrafted the wording of the ($LUE$) definition as follows: "… Another important parameter in this study

is the light use efficiency (*LUE*), which expresses the efficiency of light use in photosynthetic processes by the canopy and is defined as the ratio between *NEE* and PAR$_i$. Several other procedures have been used to approximate the *LUE*; some use the coefficient of proportionality between the *NEE* and the $PAR(D)$ (Moreira, 2017) radiation, and others use temperature measurement directly on the leaf of the trees (LI-COR) to capture the photosynthetic response as a function of the variation in light intensity (Doughty, 2010)…" (L-273-277)

**22.** L281. List the bins used?

**AC:** We indicate in the text the *SZA* ranges. We used *SZA* intervals every 25 (deg), considered statistically acceptable for the data set. We also performed analyses with smaller intervals (e.g., bins every 20 and 15 deg) but found few or no points in some bins. A similar procedure has been widely used (Gu et al., 1999, [Gu et al., 2003]; Oliveira et al., 2007; Cirino et al., 2014). (L-303-314)

**23.** L290. Elaborate and briefly describe/provide examples of what were the acceptable levels?

**AC:** We have briefly elaborated and described a few sentences with acceptable levels examples. The sentences inserted are as followed: "...We exclude unexpected maximum and minimum values for the region, e.g., values below and above 20-40°C, 40-95%, -40-40 ($\mu$ mol m$^{-2}$s$^{-1}$), 0-1000 (Wm$^{-2}$) and 0-3000 ($\mu$mol m$^{-2}$s$^{-1}$) for T$_{air}$, RH$_{air}$, $FCO_2$, SW$_i$, and PAR$_i$, respectively..." (L323-325)

**24.** L305. What is the typical pattern of tropical forests?

**AC:** We inserted some sentences and mentioned the typical pattern in the text to clarify the sentence: "...The average daily pattern of *NEE* observed in 2005-2008 (Fig.4) follows the typical pattern of tropical forests (Gu, 1999, Niyogi, 2004, Von Randow, 2004, Araujo, 2010 and Vourlitis, 2011. Fig.4 shows maximum negative fluxes average $-13.7 \pm 6.2$ $\mu$mol m$^{-2}$s$^{-1}$ around 10-11h (LT), and the maximum positive fluxes average $+6.8 \pm 5.8$ $\mu$mol m$^{-2}$s$^{-1}$ during the night period between 19h and 05h (LT). We observed a slight difference in the pattern of the daily cycle of the *NEE* between the wet and dry seasons (Fig.4), with shift (an advance) in the peak absortion of CO$_2$ from the wet-to-dry season, from about 12h (LT) to 10h (LT), respectively (Fig.4)..." (L339-342)

**25.** L326. Statistical difference of what, exactly?

**AC:** Statistical (descriptive) parameters such as mean, median, coefficient of determination (R$^2$), and standard deviation present similar values. We mention these parameters in the text to clarify the sentence. (L-332-336)

**26.** Figures 4–9. The authors often refer to top panel/bottom panel of these figures, but they are mostly side by side. I suggest labelling them with (a) and (b) and edit text accordingly. Also, in many captions, the authors could describe the figures in a bit more detail, and avoid using "correlation" as a synonym of "scatter plots".

**AC:** Thank you very much for the suggestions. As mentioned, we labeled several figures with (a) and (b) and edited the text accordingly. Moreover, we've checked and improved the captions of some figures, describing them in a bit more detail. We checked both terms "correlation" and "scatter plots" and adjusted the text accordingly.

**27.** Figure 5. The authors present the binned averages as points, but presumably each bin has a significant variability that should be acknowledged/quantified.

**AC:** We agreed! Each bin has a significant variability. We determined and quantified the variability in terms of standard deviation (STD) for each bin and put them in the captions of Figures 5a and 5b. Moreover, additional figures are presented in the supplementary material with the STD (Figures S12a and S12b). (see Tab 3).

**28.** Table 4. Last header column should be statistic.

**AC:** Corrected. We have replaced it with "statistic".

**29.** L346-352. The discussion attributes the variation and increase in PARd to aerosol dispersion, but couldn't that be partially attributed to clouds too? Presumably the solar zenith angle colours could be telling something on the seasonality, but this is not discussed in the text. Likewise, Table 4 is not really discussed, and I wonder if this is needed in the main text.

**AC:** Yes, but in minor proportion, especially because our methods take advantage of AOD AERONET (LEV20), measured under cloudless conditions. We developed an algorithm to get the clear sky solar irradiance considering long-term measures of AOD (LEV20) (as described in the 2.3.2 section), which is calibrated for BBOA aerosols but not deeper clouds. In these conditions, the parameter $f$ can be used as a sensible and suitable indicator of the entrance of aerosols, but is not a good indicator for shallow clouds (translucent). Therefore, our analyses cannot state the complete absence of clouds. Regarding the $SZA$ (colorbar), we have made a few sentences to discuss the seasonal variations and their effects on $PAR(D)$. As for Table 4, we inserted sentences in the body of the manuscript to discuss it (L-385-403).

**30.** L376. Most of this paragraph discusses LUE but no direct link with the results of this study is provided.

**AC:** We have improved the paragraph with sentences to report the direct link with the results of this study is provided. (L421-436)

**31.** L414. Alta Floresta (2 words)?

**AC:** We have checked and corrected it. It's the name of a municipality.

**32.** L415. Where do we see the standard deviation?

**AC:** The standard deviations can be found in the previous studies published by Oliveira et al. (2007) in the FLONA-Tapajós (PA), Cirino et al. (2014) in Manaus (AM), and Ji-Paraná (RO). We have cited these studies in the text. (L476-478)

**33.** L415-417. Either discuss what the readers should get from Table 5 or move it to Supplemental Materials.

**AC:** As for Table 5, we also added a brief discussion in the main text (L482-494)

**34.** L445. Consider replacing "jumped" with "declined"

**AC:** Thanks. Replaced.

**35.** L485-487. This sentence seems to contradict the text in L474-481, and the authors did not present a clear separation between cloud and aerosol effects on NEE. I suggest dropping the sentence.

**AC:** We agreed. We deleted the sentences.

**36.** Data availability. The authors should consider depositing their code to a permanent archive too. I also found the Ameriflux remark unnecessary, considering that the authors provide a DOI link with the data (although I had to remove the .2 at the end to access it).

**AC:** We agreed. In the final document, we deleted the Ameriflux remark from the Data availability section. Moreover, we checked the hyperlink in Table S5 and noticed the DOI: m5h5fw872g/1, which is promptly accessed by dataset: Cirino, Glauber; Vourlitis, George; Silva, Simone; Palácios, Rafael (2023), "Brazil-FluxMet-Stf", Mendeley Data, V1, doi: 10.17632/m5h5fw872g.1.

**Referee Comments #2**

**General comments:**

This study uses a series of satellite and ground observation to investigate the role of biomass burning aerosols on surface radiation and net ecosystem exchange in the northern part of the Mato Grosso State in Brazil. The work addresses a very interesting topic and I think brings an important contribution to existing literature in this area. However, I believe it still requires some important revisions before it can be published.

**AC:** Thank you very much for your valuable comments and suggestions. All considerations (major and minor comments and technical comments) were addressed in the revised manuscript.

**Major Comments - RC#2**

**1.** The main weakness of the study in its current form is the absence of a thorough evaluation of the methodology proposed. In particular, it is important to present and discuss the extent to which estimates of key variables (e.g., solar irradiances, relative irradiance, leaf canopy temperature) compare with other existing estimates).

**AC:** That's right. We agreed. We have dedicated a brief period at the end of the main results sections to discuss how comparable the estimated key variables are with the methods/results used by other authors. Likewise, some responses given to Referee #1 also helped to reinforce/improve the aforementioned weaknesses. For example: (1) To relative irradiance ($f$): see sentences/discussion added (L385-403) and AC #29 to referee #1; (2) To Leaf canopy temperature ($LC_T$): see brief sentences/discussion added in section 3.6 (L545-558 and section 3.6).

**2.** It is confusing in places that all the observed NEE changes are being entirely attributed to the aerosol-driven changes in diffuse radiation, while in fact they are the combined effect of several other additional key changes (e.g. temperature). Section 3.6 mentions this a bit, but in several instances in the manuscript this seems to be overlooked. Ideally it would be good to extend the methodology in order to allow isolating the temperature effect as well; however, if this proves to be very difficult, the current limitations and their implications should be clearly mentioned to avoid any confusions.

**AC:** It's a good point raised. (1) We could use atmospheric radiative transfer models (RTMs) coupled to DVMs (Dynamic Vegetation Models), (2) set up an experiment to get a wider data set, and (3) even use part of the methods applied to estimate the effects to $VPD$ and $LC_T$. However, from our standpoint, this would require slightly more complex methodological strategies outside the scope of this study. So, we added some sentences to discuss/mention the complex and non-linear relationship between photosynthesis, water vapor, and leaf canopy temperature variables that modulate photosynthesis rates (L593-598).

**Minor Comments - RC#2**

**1.** I find the statements at lines 50-52 and 57-59 incorrect, as existing studies (including some already cited here) have in fact looked into this and quantified the impact of fire aerosols on plant CO2 absorptions.

**AC:** We have checked the sentences, corrected/replaced them with the sentences below, and reinforced the novelty of the present study.

"However, little research has been done on the ecotones in the Amazon, e.g., in the Cerrado-Amazonian Forest transition, which lies within the arc of deforestation, and other biomes such as Cerrado-Caatinga, Cerrado-Atlantic Forests, and Pantanal forests. Numeric simulations have also demonstrated the impact of aerosols on GPP on the regional (Moreira et al.2013; Rap et al., 2015; Bian et al., 2021) and global scales (Mercado et al., 2009, Rap et al., 2018), but physical representations of these impacts on transition ecosystems are still lacking. To our knowledge, this is the first study with this purpose." (L52-57)

**2.** Section 3.1: Should explain a bit more where do the differences in NEE values compared other estimates come from? You list some of the differences compared to the values from Vourlitis et al. (2011), but should also add a discussion on reasons and implications of those differences.

**AC:** Scientific questions and the underlying methodological approaches are the main differences. E.g., most of the results presented by Vourlitis et al. (2011) are on a daily or monthly basis, while this research uses an hourly approach in fragments day specifics. We have made this more explicit in the revised manuscript (see section 3.1 - L348-354)).

**3.** Could you please clarify if f, as defined in Section 2.3.3. can take values larger than 1 and what do they correspond to? These seem to be mentioned within the text (e.g. Sections 3.3 and 3.5).

**AC:** This is an excellent point. Yes, $f$ can take values larger than 1, usually due to the called "cloud gap effect" (Gu et al., 1999; 2001). In general, there are multiple scatterings of solar radiation by the clouds around the study area, but still out of the pyranometer's viewing angle. However, there is still no consensus about the amounts in the literature. It has been observed $f$ of about 1.1-1.3 for the southern Amazon (Gu et al., 2001). We have added some sentences about this effect to clarify the results presented in Figure 8a (L220-225)).

**4.** Throughout the manuscript, there seems to be an assumption that biomass burning is the only aeosol species affecting surface radiation and NEE. It would be good to discuss the extent to which other aerosol species (e.g. different aerosol optical properties (e.g.single scattering albedo) affect radiation and NEE?

**AC:** We agreed! We have inserted a few sentences to clarify this discussion. BBOA is the predominant aerosol in the region, especially during the burning season, but about of 10% of the burn plume load is composed of BC (Black Carbon) and BCr (Brown Carbon), for which the Single Scattering Albedo (SSA) is affected. In general, these particles have the potential to heat the atmosphere (absorption is greater than reflection), producing values that may be above the optimal physiological thresholds of the ecosystem influencing the $NEE$ (L398-403)

**Technical Corrections - RC#2**

**1.** Line 50: "litter" should be "little".

**AC:** We replaced it with 'little'.

**2.** Line 76: What is meant by "region of 105 continuous agricultural expansion"?

**AC:** Thanks. This was a typing error.

**3.** Fig 4.: Should clarify in caption what are the NEE values illustrated based on.

**AC:** We hope this has clarified what the $NEE$ values illustrated. The revised caption is below:

$NEE$ average hourly cycle between June/2005 and July/2008, during the rainy (a) and dry (b) seasons for the semideciduous forest at the Claudia municipality. No filters are applied. The $NEE$ is presented for any sky conditions during the year. The standard deviation is shown as vertical bars.

**4.** 5-8: Unclear what is meant by top and bottom panels (mentioned both in the figure captions and within the text).

**AC:** Corrected. We removed all observed cases and replaced them with single letters (e.g., a, b, c, …) as per figure citations in the manuscript.

**5.** Line 502: "Relative irradiance" instead of "Irradiance".

**AC:** Corrected.

**6.** There are several grammar/syntax errors throughout the text that need to be corrected.

**AC:** Please accept our apologies for grammatical errors. We have made some corrections throughout the manuscript and hope to have improved the quality of the written text.

**7.** Should revise the incorrect use of ";" throughout the manuscript.

**AC:** We checked for inappropriate use of ";" throughout the text. The incorrect cases were deleted.

[revised manuscript text omitted]

---

## Author Response (AR2)

![CC BY]

**Final Response**
**egusphere-2023-684**
**Rodrigues et al. (2023)**
**simone.silva@ig.ufpa.br**

[Figure]

**Responses and actions on referee comments on the manuscript egusphere-2023-684, Enhanced net CO₂ exchange of a semi-deciduous forest in the southern Amazon due to diffuse radiation from biomass burning, by Simone Rodrigues et al., 2023.**

Date: December 19, 2023

**Referee Comments #1**

**General comments:**

I would like to thank the authors for thoroughly addressing my previous comments and suggestions, and for implementing the new analyses to mitigate the potentially confounding factors on productivity. I think the current version of the manuscript is nearly ready for acceptance, but I identified a few very minor points that I describe below that the authors may want to revise. Line numbers refer to the version of the revised manuscript with tracked changes.

**AC:** On behalf of all coauthors, I want to express my gratitude to the editor and referees for their valuable feedback, which helped improve our paper significantly. We have accepted and corrected all the minor issues pointed out. Our responses are below (see track changes highlighted in green).

**Minor points - RC#1**

**1.** L2-L3. Perhaps simplify the sentence to something like "Few studies have hitherto investigated or measured the radiative effects of aerosols on the Amazon and Cerrado biomes".

**AC:** We agree and replaced the sentence as suggested **(L2-L3)**.

**2.** L56-57. I think the authors can drop the last sentence. Instead, the authors can mention the novelty at the last paragraph of the introduction, where they describe their research objectives.

**AC:** It's a great suggestion. We've done **(L76-L77)**.

**3.** Figure 1. The caption should describe all the panels. Specifically, the authors could describe the AOD map (data source, date or average window) and the land cover map source.

**AC:** We agree and have rewritten it as follows: "Figure 1. (a) shows the average regional distribution of AOD at 550 nm extracted from Terra(Aqua) - MOD(MYD)04-3K platforms between 2000-2020 at the studied area; (b) map of South America, highlighting the political limits of the studied region (magenta); (c) localization map micrometeorological tower in the Cláudia municipality, 50 km northeast of Sinop, Mato Grosso (white point). Changes in land use and land cover are also shown by TerraBrasilis" (**see caption**).

**4.** L107. I suggest replacing "produce new" with "produce new leaves"

**AC:** We have replaced **(L106)**.

**5.** L179. The correct name is "Kelvin" (instead of "degrees Kelvin"), because K is an absolute temperature scale.

**AC:** Thanks a lot. We have corrected **(L179)**.

**6.** Figure 4. Using vertical bars to describe the standard deviation is ambiguous. I suggest describing what the box height means (presumably interquartile range) and refer to the standard deviation range as "whiskers". Also, is the central line the median or the mean?

**AC:** Thank you for the suggestion. We have inserted all the info as follows: "Figure 4. NEE average hourly cycle between June/2005 and July/2008, during the rainy (a) and dry (b) seasons for the semideciduous NEE forest at the Claudia municipality. No filters are applied. The NEE is presented for any sky conditions during the year. We used the box plot NEE to represent the distribution of $CO_2$ flux data. The vertical bars are the maximum and minimum values. The lower and upper limits of the boxes represent, respectively, the 25th and 75th percentiles, whereas the horizontal blue and red lines represent the median of the $CO_2$ flux data" **(caption)**.

**7.** L507 and throughout the edited text. I think the reasons for not analysing GPP directly are defensible, but I wonder if the correct statement should be that "we assumed that the temporal variability of GPP is similar to the temporal variability of NEE".

**AC:** We agree and replaced the sentence as suggested **(L507)**.

**8.** L545. Consider rewriting to "we noticed that the Tair variability is wider/broader than the LCT variability".

**AC:** We've replaced the sentence as suggested **(L544-L545)**.